# Anti-relapse neurons in the infralimbic cortex of rats drive relapse-suppression by drug omission cues

Amanda Laque[1], Genna L. De Ness[1], Grant E. Wagner[1], Hermina Nedelescu[1], Ayla Carroll[1], Debbie Watry[1], Tony M. Kerr[1], Eisuke Koya[2], Bruce T. Hope[3], Friedbert Weiss[1], Greg I. Elmer[4] & Nobuyoshi Suto[1]

Drug addiction is a chronic relapsing disorder of compulsive drug use. Studies of the neurobehavioral factors that promote drug relapse have yet to produce an effective treatment. Here we take a different approach and examine the factors that suppress—rather than promote—relapse. Adapting Pavlovian procedures to suppress operant drug response, we determined the anti-relapse action of environmental cues that signal drug omission (unavailability) in rats. Under laboratory conditions linked to compulsive drug use and heightened relapse risk, drug omission cues suppressed three major modes of relapse-promotion (drug-predictive cues, stress, and drug exposure) for cocaine and alcohol. This relapse-suppression is, in part, driven by omission cue-reactive neurons, which constitute small subsets of glutamatergic and GABAergic cells, in the infralimbic cortex. Future studies of such neural activity-based cellular units (neuronal ensembles/memory engram cells) for relapse-suppression can be used to identify alternate targets for addiction medicine through functional characterization of anti-relapse mechanisms.

[1] Department of Neuroscience, The Scripps Research Institute, La Jolla, CA 92037, USA. [2] Sussex Neuroscience, School of Psychology, University of Sussex, Falmer, UK. [3] Behavioral Neuroscience Branch, Intramural Research Program, National Institute on Drug Abuse, NIH/DHHS, Baltimore, MD, USA. [4] Maryland Psychiatric Research Center, Department of Psychiatry, University of Maryland School of Medicine, Baltimore, MD 21228, USA. Correspondence and requests for materials should be addressed to F.W. (email: bweiss@scripps.edu) or to G.I.E. (email: gelmer@som.umaryland.edu) or to N.S. (email: nsuto@scripps.edu)

Relapse prevention is a major goal in the treatment of drug addiction[1], as many addicts return to compulsive drug use even after a successful period of abstinence[2]. A significant amount of research has been, therefore, dedicated to determining the neurobehavioral factors that promote drug relapse. Environmental cues predictive of drug availability, along with stress and the drug itself, are recognized as key relapse-promoting factors due to their enduring ability to trigger craving in recovering addicts and reinstate extinguished drug seeking in laboratory animals[3]. A wealth of knowledge is now available describing how these factors engage distinct brain processes to promote relapse[4]. Unfortunately, anti-relapse medications designed to counter relapse-promoting brain processes have been met with limited clinical success[5,6]. Psychosocial interventions designed to counter relapse-promoting environmental cues, such as cue-exposure therapy, have also been met with limited success[7]. Hence, an alternative research strategy may prove beneficial.

Based on this premise, we aimed to determine the neurobehavioral factors that suppress—rather than promote—drug relapse and developed an omission cue-induced suppression (OCIS) procedure to serve as an animal model of relapse-suppression. Adapting Pavlovian procedures for response inhibition, such as "conditioned inhibition"[8] and "negative occasion setting"[9], we used a discriminative stimulus predictive of drug omission (unavailability) to suppress operant drug seeking in rats. The overall goal was twofold: (1) to extend previous studies, which took similar approaches, e.g., refs. [10–17], under translationally relevant laboratory conditions linked to drug addiction states and relapse risk, and (2) to establish the causality between omission cue-induced brain process and relapse-suppression in a straightforward manner.

In the present study, we tested two versions of the OCIS procedure, each unique to a specific addiction-linked condition in male rats. The first version was for rats with regular access to cocaine over a prolonged period—a condition linked to compulsive drug use[18,19]. The second version was for alcohol dependent rats undergoing acute or protracted withdrawal—conditions linked to heightened relapse risk[20,21]. We first utilized the OCIS procedure to determine the anti-relapse action of drug-omission cues against the three major modes of relapse-promotion (drug-predictive "availability" cues, stress, and drug itself) across two major classes of abused drugs (cocaine and alcohol). We then utilized the OCIS procedure as the behavioral platform to determine the brain processes that actively suppress relapse.

We previously reported[22,23] that two distinct units of neurons, both localized within the infralimbic cortex (IL; the ventral part of the medial prefrontal cortex), exert opposing environmental actions (promotion and suppression) on appetitive behavior towards non-drug rewards (glucose and saccharin). Similar functional units of prefrontal cortical neurons have been reported[24,25]. Given that the medial prefrontal cortex is implicated in decision making and impulse control in general[26,27], we hypothesized that drug-omission cues recruit (activate), in a similar manner, such functional units of neurons ("neural ensembles"[28] and/or "memory engram cells"[29]) to suppress relapse. We therefore first determined omission cue-induced neural activity in IL, using immunohistochemistry (IHC), to identify neurons reactive to cocaine or alcohol omission cues (as marked by the molecular activation marker Fos). We then characterized the phenotypic composition of these omission "cue-reactive" neurons in IL, using multiplex in situ hybridization (RNAscope®), to determine the extent of omission cue-induced neural activity across three major neural phenotypes (glutamatergic[30], GABAergic[31] and cholinergic[32] cells) of the medial prefrontal cortex.

Contradictory reports exist regarding medial prefrontal cortical regulation of drug relapse[33,34]. Disrupting neural activity in IL, using inhibitory GABA agonists, interferes not only with extinction (suppression)[12,35,36] (also see ref. [14]), but also with reinstatement (promotion)[37,38] of drug seeking. Yet other reports indicate that GABA agonists in IL do not produce a significant effect on drug seeking[11,39] (also see ref. [40]). We hypothesized that these mixed results were due to the fact that GABA agonists would have inhibited local cells irrespective of their intrinsic activity. Such neural activity, in response to behaviorally relevant stimuli, is thought to represent a unique learned association between stimuli (Pavlovian conditioning) and/or between a behavioral response and a consequence (operant conditioning)[28,29]. Non-activity-specific inhibition of IL neurons by GABA agonists likely affects multiple learned associations at once, and it would thereby result in complex behavioral responses that are potentially not relevant to the control of drug seeking and relapse. To overcome this technical issue, we utilized a neural activity-based ablation technique (Daun02 disruption[41]) to selectively disrupt drug-omission cue-reactive neurons in IL, thus unambiguously establishing the causality between omission cue-induced neural activity and relapse-suppression.

We report here that drug-omission cues can be used to suppress three major modes of relapse-promotion (drug-predictive cues, stress, and drug exposure) across two major classes of abused drugs (cocaine and alcohol) in male rats. Omission cue-reactive neurons in IL, which constitute small subsets of glutamatergic and GABAergic cells, drive this relapse-suppression. Further characterization of such anti-relapse neurons and brain mechanisms can guide the identification of alternate targets for addiction medicine.

## Results

**Relapse-suppression by cocaine omission cues**. Rats underwent the OCIS procedure for cocaine seeking (Fig. 1a). All rats were first trained (Fig. 1b: Self-administration training) to press an "active lever" for an intravenous cocaine infusion (1.0 mg/kg) presented together with a light-cue. Both active lever and light-cue thus came to predict cocaine as "cocaine availability cues". Rats quickly learned to dissociate between the active lever and an identical but "inactive" lever. The rats were then trained (Fig. 1c: Discrimination training) to recognize an olfactory cue (orange scent) as a discriminative stimulus predictive of cocaine omission ("cocaine S-"). For this, each rat repeatedly underwent two types of once daily cue-training ("No S-" and "S-") sessions over a prolonged period. Inactive lever-pressing remained minimal during both S- and No S- training (Supplementary Fig. 1a).

The results from additional groups of rats (Fig. 1d) confirmed that, consistent with previous reports[18,19], prolonged access to cocaine resulted in punishment-resistant "compulsive" cocaine intake. Here, we determined animals' resistance to cocaine-paired punishment after the Self-administration training and Discrimination training phases. Compulsivity was measured as the percentage of numbers of self-administered cocaine, each paired with electric foot-shock, against numbers of self-administered cocaine under the baseline (no shock) condition. Compared to the rats with limited (2–3 weeks) access to cocaine through Self-administration training, the rats with additional (9+ weeks) access through Discrimination training were significantly more resistant to electric foot-shock (0.8 mA, 1-s) paired with each delivery of self-administered cocaine.

The rats were then tested for relapse-suppression by cocaine S- (Discrimination tests). For this, each rat underwent two types ("No S-" and "S-") of cue-tests. Each cue-test (6 h, total) was

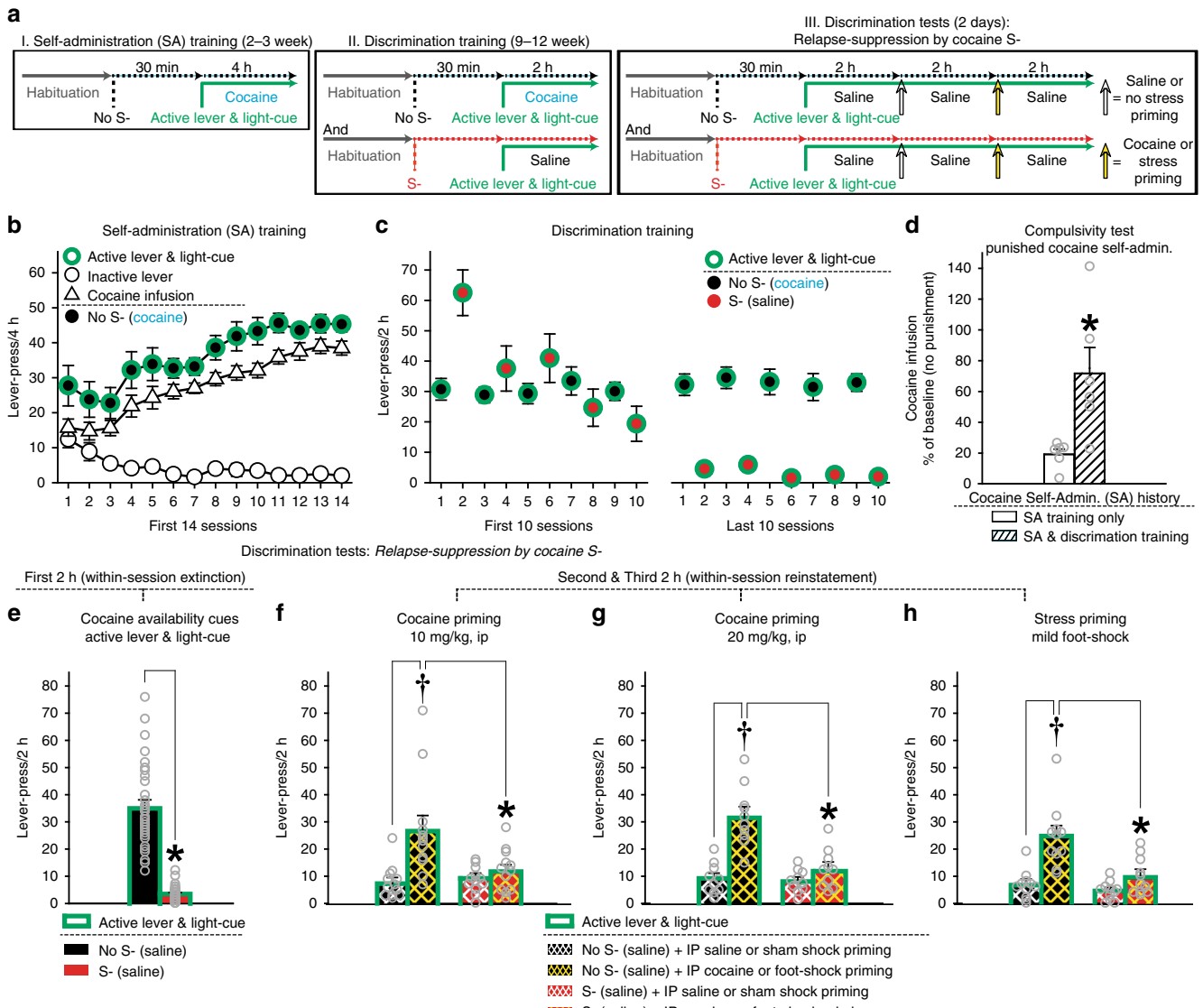

**Fig. 1** Omission cue-induced suppression (OCIS) procedure for cocaine seeking. A total of 63 rats were initially used and randomly assigned to four experimental groups: Compulsivity test, Cocaine (1.0 or 2.0 mg/kg), and Stress priming groups ($n = 14,16,16,17$). Of which, 45 rats were retained for statistical analyses (see Methods). Final Ns for each group are described below. All data are mean and SEM. Gray open circles on bar charts depict individual data-points. **a** Timeline and schedule. **b** Responses during the first 14 days of Self-administration (SA) training. **c** Responses during the first and last 10 days of Discrimination training. **d** Compulsivity tests for animals' resistance to cocaine-paired punishment after cocaine access through the SA training phase vs. both SA and Discrimination training phases. Each cocaine infusion was paired with an electric foot-shock (0.8 mA, 1-s). $n = 6,6$. Student's $t$-test: $t_{(10)} = 3.20$, *$P < 0.05$. **e** Responses during the first 2-h block of Discrimination test to determine cocaine S-'s action against cocaine availability cues (active lever and light-cue). $n = 33$. Paired $t$-test: $t_{(32)} = 11.50$, *$P < 0.001$. Duplicate data-points from multiple subjects are not overlaid on this Figure for clarity because $n = 33$. Thus, these dot plots represent the data range. **f**, **g** Responses during the second and third 2 h-blocks of Discrimination test (within-session reinstatement) to determine cocaine S-'s action against cocaine priming (10 and 20 mg/kg, IP). Two-way repeated measures ANOVA for the 10 mg/kg group ($n = 12$): Priming ($F_{(1,11)} = 19.20$, $P < 0.001$) main effects and Cue-Test x Priming interaction ($F_{(1,11)} = 10.34$, $P < 0.01$). Two-way repeated measures ANOVA for the 20 mg/kg group ($n = 10$): Cue-Test ($F_{(1,9)} = 42.02$, $P < 0.001$) and Priming ($F_{(1,9)} = 47.19$, $P < 0.001$) main effects, and Cue-Test x Priming interaction ($F_{(1,9)} = 94.49$, $P < 0.001$). †$P < 0.001$ vs. IP saline. *$P < 0.01$–0.05 vs. No S-. Tukey HSD test. **h** Responses during the second and third 2 h-blocks of Discrimination test to determine cocaine S-'s action against stress priming (mild electric foot-shock, intermittent, 10 min, 0.8 mA). $n = 11$. Two-way repeated measures ANOVA: Cue-Test ($F_{(1,10)} = 17.41$, $P < 0.01$) and Priming ($F_{(1,10)} = 121.79$, $P < 0.001$) main effects, and Cue-Test x Priming interaction ($F_{(1,10)} = 15.36$, $P < 0.01$). †$P < 0.01$ vs. sham foot-shock. *$P < 0.05$ vs. No S-. Tukey HSD test

divided into three 2-h blocks. Under both No S- and S- conditions, cocaine was not available for self-administration. Relapse-suppression against cocaine availability cues (active lever and light-cue) was determined in all rats during the first 2-h block. Relapse-suppression against cocaine and stress priming was determined in different groups of rats during the second and third 2-h blocks.

Despite the extensive drug history linked to compulsive cocaine intake[18,19] (Fig. 1d), cocaine S- suppressed the relapse-promoting action of cocaine availability cues (Fig. 1e). During the first 2-h block of the No S- test, the rats initiated and maintained active lever-pressing, even though this behavior only resulted in the light-cue but not cocaine. In contrast, during the S- test, the same rats minimally lever-pressed under otherwise identical stimulus

conditions. Thus, active lever and light-cue, as cocaine availability cues, sufficiently (without the primary cocaine reward) promoted operant drug response (cocaine seeking) in the absence (No S-)—but not the presence (S-)—of cocaine omission cues. Inactive lever-pressing remained minimal during both S- and No S- tests (Supplementary Fig. 1b).

Cocaine S- also suppressed the relapse-promoting action of cocaine (Fig. 1f, g) and stress (Fig. 1h) priming. During the No S- test, active lever-pressing for the light-cue decreased over the initial 2-h period of the 6-h test (within-session extinction). Priming by saline (0.5 ml, IP) or sham shock—given at 2 h into the 6-h test—minimally affected the then-extinguished lever-pressing. In contrast, priming by cocaine (10 or 20 mg/kg, IP) or electric foot-shock stress (0.8 mA for 0.5-s delivered over 10 min on a variable time schedule at a mean interval of 40 s), adapted from ref. [42] given 2 h later at 4 h into the 6-h test, significantly increased this cocaine-seeking behavior (within-session reinstatement). However, during the S- test, cocaine or foot-shock priming failed to reinstate the extinguished cocaine seeking. Inactive lever-pressing remained minimal during both S- and No S- tests (Supplementary Fig. 1c, d, e).

These results extend previous reports[10,11,13,14,43] and establish the anti-relapse action of cocaine omission cues.

**Relapse-suppression by alcohol omission cues.** Rats underwent the OCIS procedure for alcohol (Fig. 2a). Each rat was tested for relapse-suppression three times under different alcohol withdrawal states: non-withdrawal (NW), acute withdrawal (AW), and protracted withdrawal (PW). All rats were first trained (Fig. 2b: self-administration training) to press an "active lever" for alcohol (20%, v/v, oral) presented together with a "light-cue". Both active lever and light-cue thus came to predict alcohol as "availability cues". Rats quickly learned to dissociate between the active lever and an identical but "inactive" lever. The rats were then trained (Fig. 2c: Discrimination training NW) to recognize an olfactory cue (orange scent) as a discriminative stimulus signaling alcohol omission ("alcohol S-"). For this, each pre-dependent rat repeatedly underwent two types of once daily cue-training ("No S-" and "S-") sessions. Inactive lever-pressing remained minimal during both S- and No S- training (Supplementary Fig. 2a).

The rats were then tested for relapse-suppression by alcohol S- (Discrimination tests NW) before being subjected to an experimental procedure to induce alcohol dependence (alcohol liquid diet, adapted from refs. [43,45]). The rats were further trained (Fig. 2d) for alcohol S- during acute (5–8 h) withdrawal from alcohol (Discrimination training AW). Alcohol liquid diet was initially available continuously for 3 weeks, and then intermittently (14 h daily) to induce AW. Operant training was resumed 3–4 days after this schedule change. Consistent with previous reports[44,45], AW from alcohol liquid diet resulted in a small but significant increase in operant responding for alcohol (Fig. 2c, d). Similar alcohol diet schedules are known to produce somatic and emotional withdrawal symptoms in rats[46,47]. The observed increase in operant responding, evident from the initial No S- (alcohol) sessions under AW (as in refs. [45,48]), was thus presumably because the rats had learned to alleviate such symptoms by consuming alcohol (initially in alcohol liquid diet). Inactive lever-pressing remained minimal during both S- and No S- training while rats undergoing AW (Supplementary Fig. 2b). Lastly, the rats were tested for relapse-suppression by alcohol S- during acute (5–8 h) and protracted (2+ weeks) withdrawal from alcohol (Discrimination tests AW and PW).

During Discrimination tests NW, AW, and PW, each rat underwent two types ("No S-" and "S-") of cue-tests. Under both

No S- and S- conditions, alcohol was not available for self-administration. Each cue-test (3 h, total) was divided into three 1-h blocks. Relapse-suppression against alcohol availability cues (active lever and light-cue) was determined in all rats during the first 1-h block. Relapse-suppression against alcohol and stress priming was determined in different groups of rats during the second and third 1-h blocks.

Consistent with previous reports, e.g., ref. [49], when primed by alcohol availability cues, stress or alcohol, the rats pressed the active lever significantly more during Discrimination tests AW and PW than NW (Fig. 2e, f, g). Such enhanced responding is presumably due to alcohol withdrawal linked to heightened relapse risk[20,21], although it may also result from the additional operant training for alcohol through Discrimination training AW. Regardless of the mechanisms, alcohol S- suppressed the relapse-promoting action of alcohol availability cues in rats subjected to alcohol liquid diet (Fig. 2e). During the No S- test, the rats engaged in active lever-pressing, even though this behavior only resulted in the light-cue but not alcohol. In contrast, during the S- test, the same rats minimally lever-pressed under otherwise identical stimulus conditions (active lever and light-cue but no alcohol). Thus, active lever and light cue, as alcohol availability cues, sufficiently (without the primary alcohol reward) promoted operant drug response (alcohol seeking) in the absence (No S-)—but not the presence (S-)—of alcohol omission cues. Inactive lever-pressing remained minimal during both S- and No S- tests under the NW, AW, and PW conditions (Supplementary Fig. 2c).

Furthermore, alcohol S- significantly suppressed the relapse-promoting action of alcohol (Fig. 2f) and stress (Fig. 2g) priming. During the No S- test, active lever-pressing for light-cue decreased over the initial 1-h period of the 3-h test (within-session extinction). Priming by water (0.2 ml, oral) or saline (0.5 ml, IP)—given at 1 h into the 3-h test—minimally affected the then-extinguished lever-pressing. In contrast, priming by alcohol (20%, v/v, 0.2 ml, oral), adapted from ref. [50] or the pharmacological stressor yohimbine (0.75 mg/kg, IP), adapted from ref. [51] given 1 h later at 2 h into the 3-h test, significantly increased this alcohol-seeking behavior (within-session reinstatement). However, during the S- test, neither alcohol nor yohimbine resulted in reinstatement. Inactive lever-pressing remained minimal during both S- and No S- tests under the NW, AW, and PW conditions (Supplementary Fig. 2d, e).

These results extend previous reports[16,17] and establish the anti-relapse action of alcohol omission cues. While the pharmacological action of alcohol and cocaine are different, as in the case of the relapse-promoting cues, a similar learning mechanism thus appears to mediate the environmental control of alcohol and cocaine seeking.

**Drug-omission cue-reactive neurons in the infralimbic cortex.** Two groups of rats were trained under the OCIS procedures to learn an odor cue (orange scent) as either cocaine S- or alcohol S- (Fig. 3a, b). The rats trained for alcohol S- were not subjected to alcohol liquid diet, and thus trained and tested under a non-withdrawal state; this was done to isolate neural activity due to omission cues from neural activity due to alcohol withdrawal (e.g., ref. [52]). Another group of rats was trained as the olfactory control and exposed to the same odor without a scheduled consequence (i.e., odor-habituation: Fig. 3c). For each training condition, the rats were further randomly divided into two groups defined by the final cue-condition and exposed to the odor ("S-") or no odor ("No S-") for 90 min for optimal induction of Fos, then deeply anesthetized and euthanized. Active lever, light-cue, and drug reward were all withheld to determine neurons

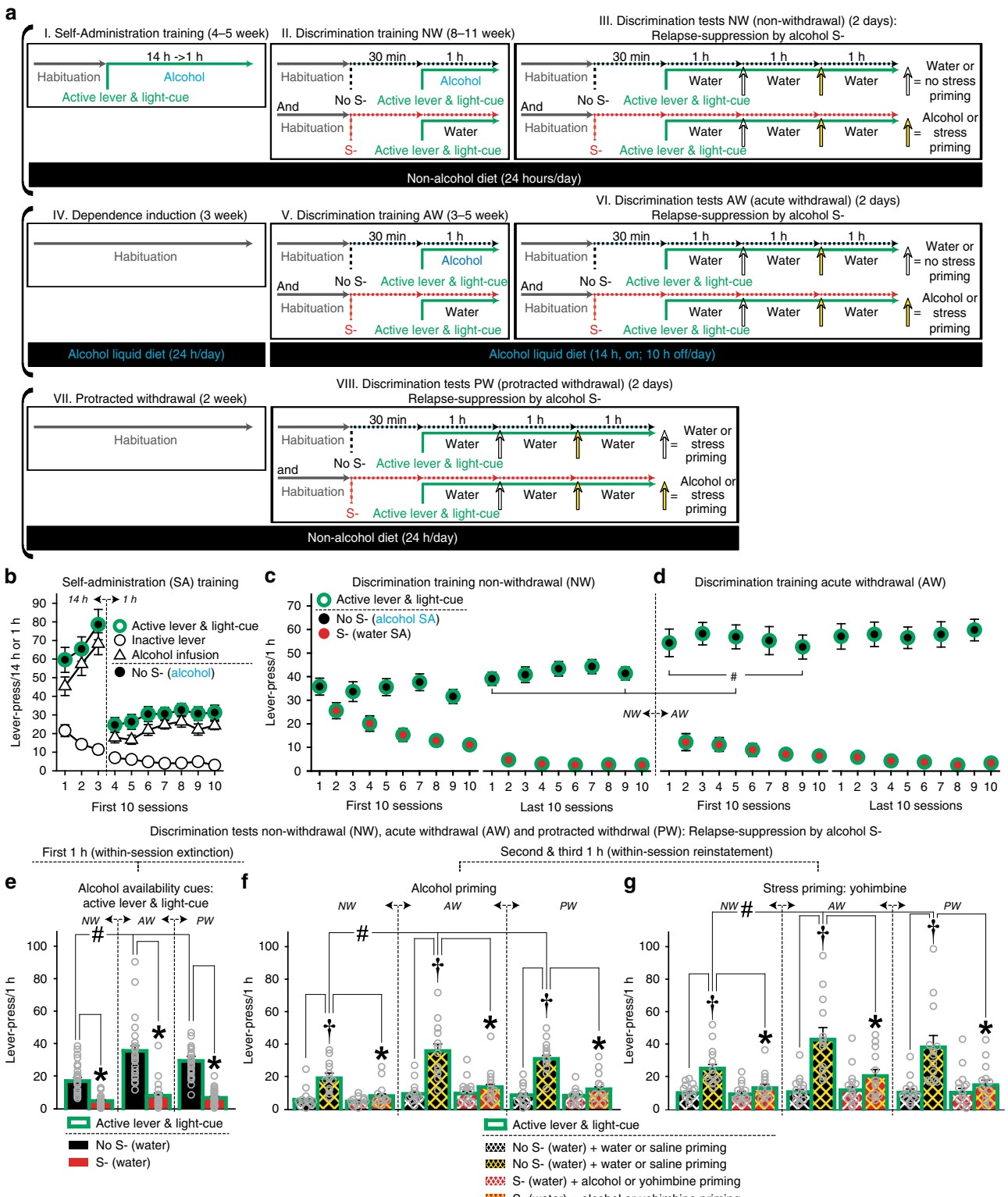

specifically reactive to S- (or No S-). Their brains were harvested and processed for Fos IHC (Fig. 3d, e) to determine omission cue-triggered neural activation in IL.

Extending the previous reports[11,15,22], the odor cue (vs. no odor cue), conditioned as cocaine S- or alcohol S-, significantly increased the numbers of Fos-positive (Fos+) cells in IL (Fig. 3g). In contrast, an identical but non-conditioned and well-habituated odor cue did not. The observed neural activity in IL (Fig. 3g) is thus presumably due to the omission (learning) rather than

olfactory (sensory) property of the S- odor cue. Interestingly, the odor cue increased Fos+ nuclei to similar extents in different groups of rats trained for cocaine S- and alcohol S-, suggesting that similar neuronal processes mediate the anti-relapse action of S- across different classes of abused drugs.

To further examine this possibility, additional groups of rats were trained for the same odor cue as cocaine S- (Fig. 3a) or alcohol S- (Fig. 3b). The rats were exposed to the S- odor for 90 min, then deeply anesthetized and euthanized. Active lever, light-

**Fig. 2** Omission cue-induced suppression (OCIS) procedure for alcohol seeking. All data are mean and SEM. Gray open circles depict individual data-points. Thirty-three rats were initially used and randomly assigned to alcohol and stress priming groups ($n = 15, 18$). Of which, 25 rats were retained for statistical analyses (see Methods). Final Ns per group are described below. **a** Timeline and schedule. **b** Responses during the first 10-days of Self-administration (SA) training. **c, d** Responses during the last and first 10 days of Discrimination training under non-withdrawal (NW) and acute withdrawal (AW) conditions. Operant training was resumed 4–5 days following the diet schedule change to induce AW. $n = 25$. Two-way repeated measures ANOVA: Withdrawal ($F_{(1,24)} = 18.27$, $P < 0.001$) main effects. #$P < 0.01$–$0.05$ vs. NW. Tuckey HSD test. **e** Responses during the first 1-h block of Discrimination test to determine the anti-relapse action of alcohol S- against alcohol availability cues under NW, AW, and protracted withdrawal (PW). $n = 25$. Two-way repeated measures ANOVA: Withdrawal ($F_{(2,48)} = 15.01$, $P < 0.001$) and Cue-Test ($F_{(1,24)} = 120.78$, $P < 0.001$) main effects, Withdrawal x Cue-Test interaction ($F_{(2,48)} = 11.85$, $P < 0.001$). #$P < 0.01$–$0.05$ vs. NW. *$P < 0.001$ vs. No S-. Tuckey HSD test. Duplicate data-points from multiple subjects are not overlaid on this Figure for clarity because $n = 25$. Thus, the dot plots represent the data range. **f** Responses during the second and third 1-h blocks of Discrimination test to determine alcohol S-'s action against alcohol priming under NW, AW, and PW. $n = 12$. Three-way repeated measures ANOVA: Withdrawal ($F_{(2,22)} = 13.22$, $P < 0.001$) and Priming ($F_{(1,11)} = 97.50$, $P < 0.001$) main effects, and Withdrawal x Priming ($F_{(2,22)} = 16.48$, $P < 0.001$), Cue-Test x Priming ($F_{(1,11)} = 24.22$, $P < 0.001$) and Withdrawal x Cue-Test x Priming ($F_{(2,22)} = 14.78$, $P < 0.001$) interactions. #$P < 0.01$–$0.05$ vs. NW. †$P < 0.01$ vs. water. *$P < 0.01$–$0.05$ vs. No S-. Bonferroni test. **g** Responses during the second and third 1-h blocks of Discrimination test to determine alcohol S-'s action against stress priming under NW, AW, and PW. $n = 13$. Three-way repeated measures ANOVA: Priming ($F_{(1,12)} = 27.65$, $P < 0.001$), Withdrawal x Priming ($F_{(2,24)} = 8.34$, $P < 0.01$), Cue-Test x Priming ($F_{(1,12)} = 17.70$, $P < 0.01$), and Withdrawal x Cue-Test x Priming ($F_{(2,24)} = 6.20$, $P < 0.01$) interactions. #$P < 0.05$ vs. NW. †$P < 0.01$ vs. saline. *$P < 0.05$ vs. No S-. Bonferroni test

cue, and drug reward were all withheld to determine neurons specifically reactive to S-. Their brains were harvested and processed for RNAscope® (Fig. 3f) to determine omission cue-activation across major cortical neural phenotypes. For this, each IL section was probed for four types of messenger RNA (mRNA) (encoding protein): c-fos (Fos), Slc17a7 (vesicular glutamate transporter 1 [VGLUT1]), Slc32a1 (vesicular gamma-aminobutyric acid transporter [VGAT]), and CHAT (choline acetyltransferase [ChAT]), each as a marker for omission cue-activated (S- reactive), glutamatergic (GLU), GABAergic (GABA), and cholinergic (ACh) nuclei. Each nucleus was identified by DNA-staining 4′,6-diamidino-2-phenylindole (DAPI) and used as the "region of interest" (ROI) for phenotype identification. On average (± SEM), we analyzed 1946.2 (± 200.1) nuclei per animal and conducted three lines of analyses.

First, we determined the overall extent of omission cue-activation as well as the overall phenotypic compositions—independent of S- reactivity—in IL (Fig. 3h). Consistent with the results from Fos IHC (Fig. 3g), both cocaine S- and alcohol S-induced similar levels of neural activation, as indicated by the numbers of S- reactive (c-fos+) cells, in IL. The significant phenotypic majority in IL, independent of neural activity, is GLU cells (~45%) with a small population of GABA cells (~5%) and a miniscule population of ACh cells (~1%). The majority of the remaining ("other") cells are presumably non-neural glial cells (~51%). Second, we determined the extent of omission cue-activation within each phenotype (Fig. 3i). Both cocaine S- and alcohol S- induced similar levels of neural activation in GLU (8–10%), GABA (9–13%) and other (3–5%) cell types, but did not induce any detectable activation in ACh cells (0%). Third, we determined the phenotypic composition of omission cue-activated IL neurons (Fig. 3j). IL neurons reactive to cocaine S- or alcohol S- had similar phenotypic compositions: GLU (90–94%), GABA (4–7%), and other (2–3%)—but not ACh (0%)—cell types.

We also determined the extent of omission cue-activation, as indicated by the numbers of S- reactive (c-fos+) nuclei, across cortical layers I, II, III, and V/VI, based on ref. [53] (layer V and VI were analyzed jointly; no layer IV in rats). This additional analysis (Supplementary Fig. 3a, b, c) revealed that IL neurons reactive to either cocaine S- or alcohol S- were present at similar levels across layers II, III, and V/VI, but were sparse in layer I, known to contain few neurons.

In summary, both cocaine S- and alcohol S- induced similar degrees of neural activation in IL with similar phenotypic composites—suggesting an overlapping anti-relapse mechanism

in this site. Thus, we next tested the hypothesis that drug-omission cues suppress relapse by recruiting functional cellular units of S- reactive neurons (neural ensembles and/or memory engram cells) in IL.

**Anti-relapse neurons in the infralimbic cortex.** Fos-lacZ transgenic rats[54] were trained under the OCIS procedure to learn an olfactory cue (orange scent) as cocaine S- (Fig. 4a). The rats were then randomly divided into four groups ("Group"), defined by disruption-cue ("S-" or "No S-") and microinjection ("Daun02" or "vehicle") for neural activity-targeted inactivation[41]. Each rat was first exposed to either S- or No S- for 90 min and then received a bilateral microinjection of Daun02 (2.0 μg/0.5 μl/side) or vehicle (0.5 μl/side) into IL (Fig. 4b). Active lever, light-cue and cocaine were all withheld to target neurons specifically reactive to S- (or No S-). In Fos-lacZ rats, Daun02 (inactive compound) is catalyzed into daunorubicin (cytotoxin) by beta-galactosidase (enzyme) only in Fos+ "activated" cells, thereby triggering apoptosis[40]. In contrast, Daun02 cannot be catalyzed into daunorubicin in "non-activated" cells lacking Fos/beta-galactosidase, and no cellular disruption occurs.

Each rat then underwent the Discrimination tests to determine the effects of neural activity-based ablation of omission cue-reactive neurons on relapse-suppression (Fig. 4c, d). In the two vehicle-treated groups ("No S- & vehicle" and "S- & vehicle"), cocaine S- suppressed relapse-promotion by cocaine availability cues (active lever and light-cue) and cocaine priming (20 mg/kg, IP). In the group that received Daun02 without being exposed to cocaine S- ("No S- & Daun02"), cocaine S- also suppressed such relapse-promotion, and thus its anti-relapse action was preserved. In contrast, in the group that received Daun02 following exposure to cocaine S- ("S- & Daun02"), cocaine S- failed to suppress relapse-promotion by cocaine availability cues and cocaine priming. Thus, Daun02 disruption of omission cue-reactive neurons in IL blocked the anti-relapse action of cocaine S-. For all groups, inactive lever-pressing remained minimal during both S- and No S- tests (Supplementary Fig. 4a, b).

For all cases, Daun02 disruption was verified by Fos IHC (Fig. 4e). For this, all rats were exposed to S- for 90 min, deeply anesthetized and euthanized. Active lever, light-cue and cocaine were all withheld to determine the neurons specifically reactive to S-. Compared to the "No S- & vehicle", "S- & vehicle", and "No S- & Daun02" groups, the "S- & Daun02" group exhibited significantly reduced numbers of Fos+ nuclei in IL. The fact that the number of the remaining Fos+ cells in this group ($25.5 \pm 3.7$ nuclei per mm²) is nearly identical to the corresponding number

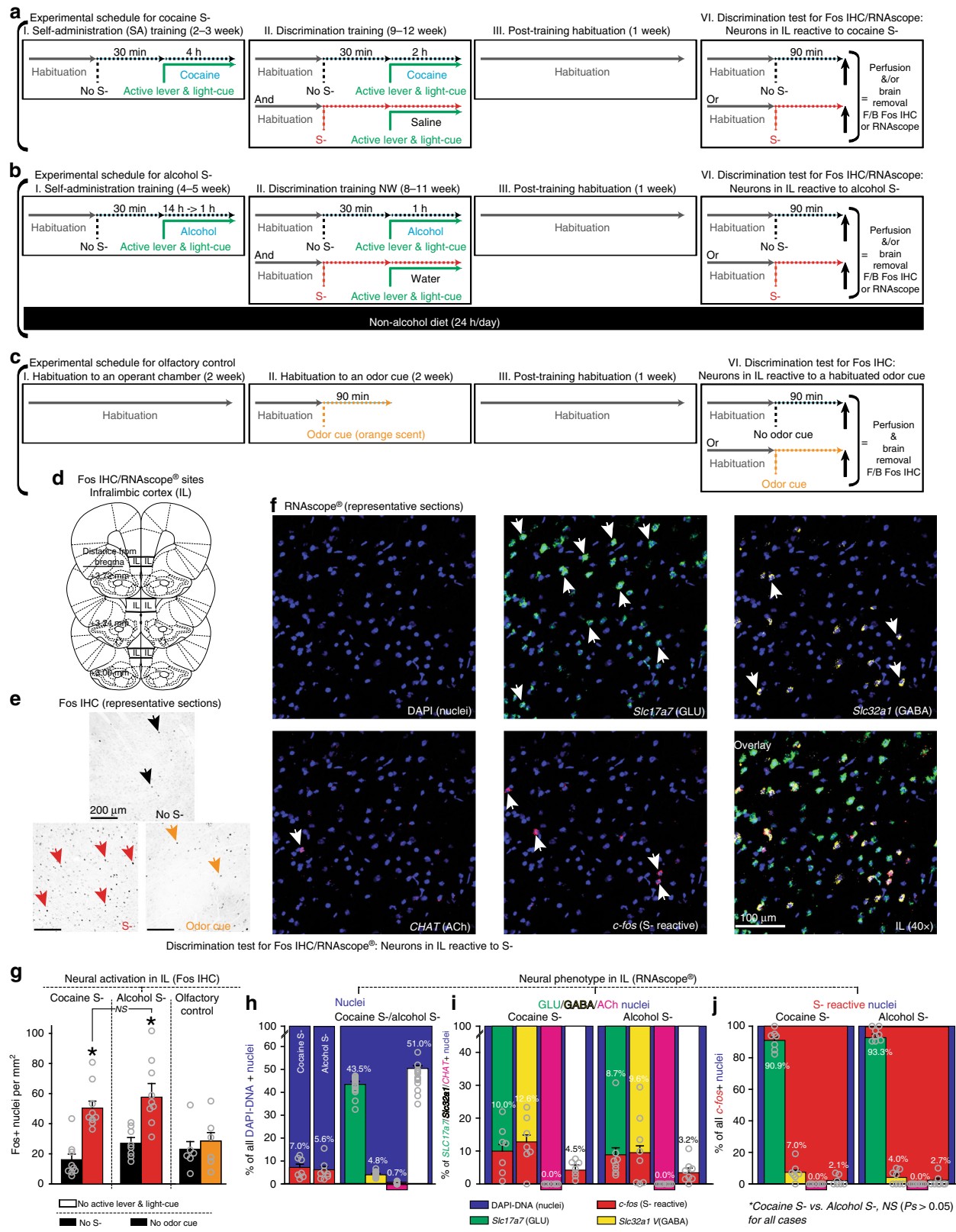

in the "no odor cue" controls ($23.2 \pm 5.1$ nuclei per mm²: see Fig. 3g) suggests that these remaining Fos+ cells likely represent a random set of spontaneously active cortical neurons.

In summary, activity-based ablation of omission cue-reactive IL neurons by Daun02 prevented relapse-suppression.

## Discussion

Under the OCIS procedure, drug-omission cues suppressed the three major modes of relapse-promotion (drug-predictive cues, stress and drug priming)[3] across two major classes of abused drugs (cocaine and alcohol) in male rats. These anti-relapse

**Fig. 3** Omission cue-induced suppression (OCIS) procedures for localization and phenotypic characterization of omission cue-reactive neurons in IL. All data are mean and SEM. Gray open circles on bar charts depict individual data-points. **a, b, c** Timeline and schedule. **d** Target sites (line drawings adapted from Paxinos and Watson[81] with permission). **e, f** Representative sections. **g** Effects of cocaine S-, alcohol S-, and well-habituated odor on neural activation in IL as indicated by Fos immunohistochemistry. $n = 10,9,8,9,6,6$. Two-way between-subjects ANOVA: Training ($F_{(2,42)} = 4.81$, $P < 0.05$) and Cue-Test ($F_{(1,42)} = 28.67$, $P < 0.001$) main effects, and Training x Cue-Test interaction ($F_{(2,42)} = 3.77$, $P < 0.05$). *$P < 0.001$ vs. No S-. Tuckey HSD test. **h, i, j** Neural phenotypes in IL reactive to cocaine or alcohol S- as indicated by in situ hybridization via 4-plex RNAscope® targeting *c-fos*, *Slc17a7*, *Slc32a1*, and *CHAT*, as markers for "S- reactive", "glutamatergic (GLU)", "GABAergic (GABA)", and "cholinergic (ACh)" nuclei. Each nucleus was identified by DAPI. For statistical analyses, total numbers of nuclei per mm$^2$ that satisfied each phenotypic criterion were used. For graphic representations, percentages of each phenotype within a specific "parent" phenotype were used. **h** Percentages of different phenotypes within all DAPI-positive nuclei. $n = 7,8,15$. Individual data-points are not overlaid on the right panel for clarity because $n = 15$. For this panel, data from rats tested for cocaine S- and alcohol S- were pooled to represent the overall percentages of different phenotypes independent of neural activity. Two-way mixed ANOVA: Phenotype ($F_{(4,52)} = 532.79$, $P < 0.001$), but not Group ($F_{(1,13)} = 4.05$, NS) or Group x Phenotype interaction ($F_{(4,52)} = 0.34$, NS). $n = 7,8$. **i** Percentages of S- reactive nuclei within different phenotypes. Two-way mixed ANOVA: Phenotype ($F_{(3,39)} = 38.62$, $P < 0.001$), but not Group ($F_{(1,13)} = 2.5$, NS) or Group x Phenotype interaction ($F_{(3,39)} = 1.74$, NS). **j** Percentages of different neural phenotypes within S- reactive nuclei. Two-way mixed ANOVA: Phenotype ($F_{(3,39)} = 27.77$, $P < 0.001$), but not Group ($F_{(1,13)} = 2.20$, NS) or Group x Phenotype interaction ($F_{(3,39)} = 2.04$, NS). $n = 7,8$

actions were observed under laboratory conditions (extensive drug access and withdrawal) linked to compulsive drug use[18,19] and heightened relapse risk[20,21]. The present results thus support the translational relevance of the OCIS procedures as an animal model to investigate the neurobehavioral processes against relapse.

The anti-relapse action of drug omission cues (also see refs. [10,11,13,14,17,43]) is presumably mediated by a learning process similar to "conditioned inhibition"[8] or "negative occasion setting"[9] and is distinct from "extinction" of operant responding for relapse-promoting availability cues. Consistent with this presumption, the OCIS procedure was effective against stress (footshock and yohimbine) and drug priming (cocaine and alcohol) known to reinstate drug seeking even after extensive extinction training[3]. Similar actions of drug omission cues have been reported[11,17,43]. Reminiscent of these differences between OCIS and extinction, omission cues without a history of strong excitatory conditioning (such as the odor stimulus in the current study) are reportedly capable of suppressing unconditioned responses (e.g., fear response triggered by brain electric stimulation); whereas, omission cues with a history of strong excitatory conditioning (such as an extinguished drug-paired context) are not[55,56].

While response inhibition through extinction is environmental context-specific[57], response inhibition through similar omission cue procedures for non-drug rewards can be preserved in different contexts[58]. Thus, the anti-relapse action of drug omission cues may transcend the context-specificity and major limitation of the cue-exposure therapies[7] designed to extinguish relapse-promoting cues through extinction training. Indeed, omission cue-training has been applied in humans to suppress learned responses, such as conditioned fear response[59–61]. However, the clinical application of the OCIS procedure is likely limited due to the difficulty of contrasting drug access with no-access conditions (discrimination training) extensively.

Nevertheless, the OCIS procedure provides a behavioral platform to study the brain processes that actively suppress—rather than promote—drug seeking, craving, and relapse. A particular advantage of OCIS over extinction is that the inhibition of learned drug responses is controlled by a single cue, uniquely associated with drug omission. In contrast, the response inhibition through extinction is controlled by an extinguished drug-paired context, the composite of multiple drug omission cues, which originally signaled drug availability. Unlike such extinction incorporated into standard reinstatement procedures, the extinction under the "ABA renewal" procedure[57,62] shares similarities with OCIS. This type of extinction is established in a strictly non-drug-paired—thus "omission"—context ("Context

B"), which is distinct from the context ("Context A") in which operant drug response is initially established. Such omission context has been shown to block the relapse-promoting action of drug (alcohol) priming[16]. However, several procedural differences exist between the ABA renewal and OCIS, such as the use of (1) singular (ABA) vs. repetitive (OCIS) contrasting between drug availability and drug omission conditions and (2) compound (ABA) vs. simple (OCIS) stimuli to signal drug omission. Whether such differences influence relapse-suppression and corresponding brain processes still needs to be elucidated. Finally, unlike under the "conditioned suppression" (e.g., ref. [19]) or "punishment" (e.g., ref. [18]) procedures, the conditioned stimulus to inhibit learned responses is not coupled with an aversive event (e.g., electric foot-shock) under the OCIS procedure. In summary, the relative simplicity of the OCIS procedures makes it more straightforward to determine the relationship between a specific brain process and relapse-suppression.

The OCIS of drug seeking is, in part, mediated by omission cue-reactive neurons in IL. In this brain region implicated in decision making and impulse control[26,27], cocaine and alcohol omission cues induced similar levels of neural activation and recruited neurons with similar phenotypic characteristics. While addiction to cocaine and alcohol are marked by some overlapping but also each distinct mechanisms[63], a similar neuronal process may therefore underlie learned suppression of drug seeking across different drug classes.

The observed neural activation in IL is likely caused by omission cue-triggered release of glutamate, which is the primary source of cortical excitation[64]. Glutamatergic inputs to IL originate from diverse brain sites[65], including the hippocampus, the thalamus, the hypothalamus, and the amygdala. Additionally, IL receives cholinergic inputs, which also excite cortical neurons, from the basal forebrain and the brainstem[65]. Other neurotransmitters (e.g., dopamine and serotonin) and peptides (e.g., endogenous opioids and neurotensin) in IL[66] can also modulate cortical excitability. Thus, OCIS may be processed through an ensemble of multiple afferent circuits and neurochemicals, rather than a specific projection or transmitter.

Within IL, omission cue-reactive neurons were concentrated in layers II, III, and V/VI, but were sparse in layer I. The predominant omission cue-reactive neural phenotype was glutamatergic and thus presumably cortical pyramidal neurons. These excitatory cells constitute the majority of cortical neurons[30] and project to other cortical and subcortical sites[67]. Within the glutamatergic population, sizable subsets (~10%) were omission cue-reactive; thus, OCIS may be processed through an ensemble of multiple omission cue-reactive efferent circuits, rather than a singular projection. Sizable subsets

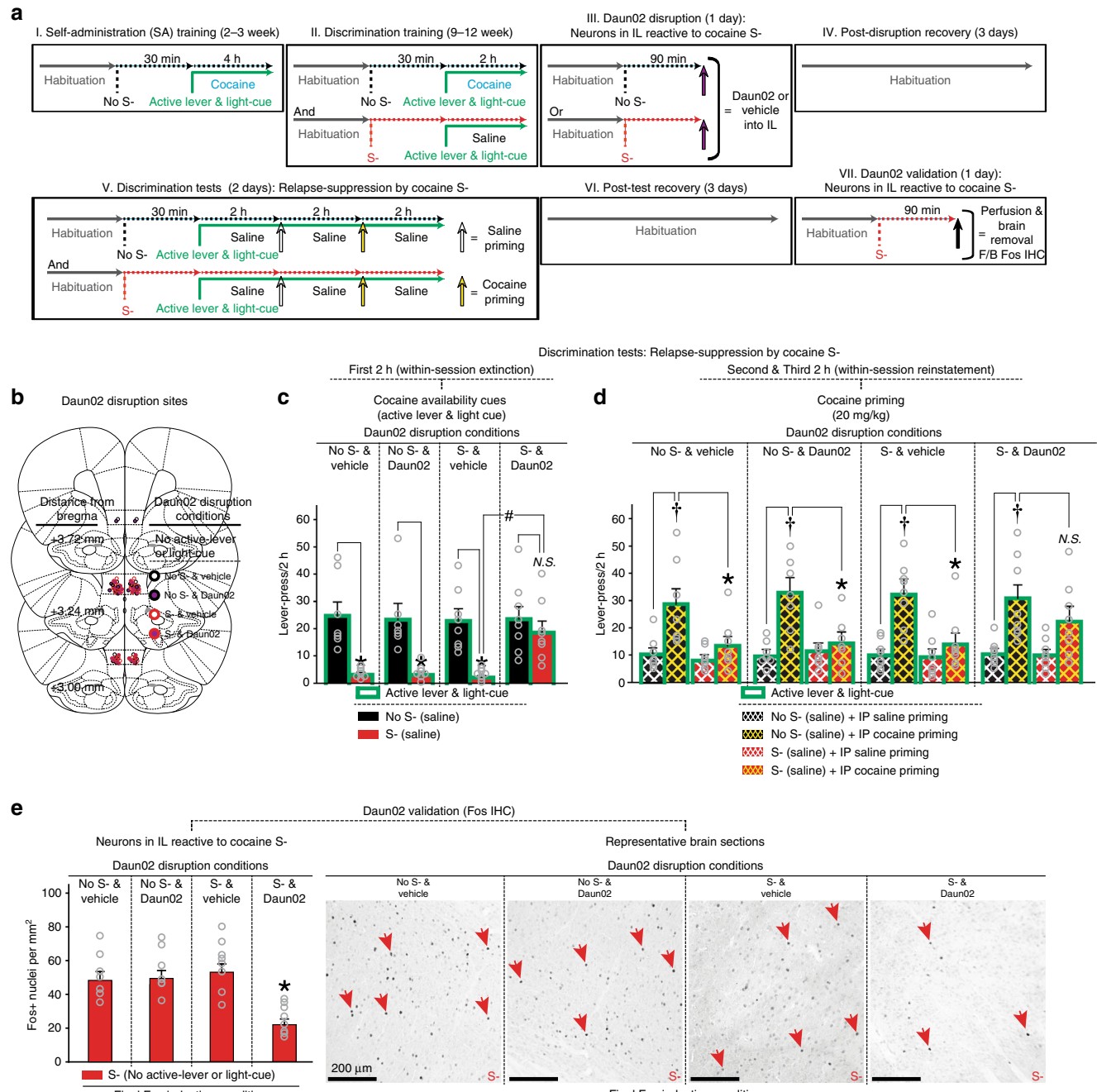

**Fig. 4** Omission cue-induced suppression (OCIS) procedures for functional characterization of omission cue-activated neurons via activity-based Daun02 disruption. All data are mean and SEM. Gray open circles on bar charts depict individual data-points. Tukey HSD test was used for all post-hoc analyses. **a** Timeline and schedule. **b** Daun02 disruption sites (line drawings adapted from Paxinos and Watson[81] with permission). **c** Responses during the first 2-h block of Discrimination test to determine cocaine S-'s action against cocaine availability cues. $n = 7,7,8,8$. Two-way mixed ANOVA: Cue-Test ($F_{(1,26)} = 37.70$, $P < 0.01$) main effects and Group x Cue-Test interaction ($F_{(3,26)} = 3.49$, $P < 0.05$).*$P < 0.001$ vs. No S-. #$P < 0.05$. **d** Responses during the second and third 2-h blocks of Discrimination test to determine anti-relapse action of cocaine S- against cocaine priming. Two-way repeated measures ANOVA was separately conducted for each group. "No S- & vehicle" group ($n = 7$): Priming ($F_{(1,6)} = 7.50$, $P < 0.05$) main effects and Cue-Test x Priming interaction ($F_{(1,6)} = 12.88$, $P < 0.05$). "No S- & Daun02" group ($n = 7$): Cue-Test ($F_{(1,6)} = 8.04$, $P < 0.05$) and Priming ($F_{(1,6)} = 25.53$, $P < 0.01$) main effects, and Cue-Test x Priming interaction ($F_{(1,6)} = 11.27$, $P < 0.05$). "S- & vehicle" group ($n = 8$): Priming ($F_{(1,7)} = 21.91$, $P < 0.01$) main effects and Cue-Test x Priming interaction ($F_{(1,7)} = 17.31$, $P < 0.01$). "S- & Daun02" group: Priming ($F_{(1,7)} = 19.50$, $P < 0.01$) main effects but not Cue-Test ($F_{(1,7)} = 3.84$, NS) or Cue-Test x Priming interaction ($F_{(1,7)} = 1.71$, NS). †$P < 0.001$ vs. saline. *$P < 0.01$–0.05 vs. No S-. **e** Daun02 validation via Fos IHC. $n = 7,7,8,8$. One-way between-subject ANOVA: Group ($F_{(3,26)} = 10.25$, $P < 0.01$). *$P < 0.05$ vs. all other groups

(~13%) of GABAergic cells were also omission cue-reactive. Presumably, these are inhibitory interneurons, which form local circuits and gate cortical signal flows[31]. Small numbers of cholinergic cells, presumably interneurons[32], were

also present but not recruited by either cocaine or alcohol omission cues.

The relative contribution of each omission cue-reactive phenotype still needs to be determined. It is, however, tempting to

speculate that the inhibitory control of drug seeking is orchestrated by excitatory efferent-ensembles of IL pyramidal cells, which project to brain sites thought to regulate learned responses, such as the basolateral amygdala[68] and the nucleus accumbens shell[69]. Meanwhile, inhibitory local circuit-ensembles of GABAergic interneurons may have disrupted IL neurons that normally promote relapse[37]. Unfortunately, no existing technique allows for separate manipulation of neurons with different phenotypic characteristics (e.g., glutamatergic vs. GABAergic) based on their intrinsic activity (e.g., cue-reactive vs. non-reactive) in rats. Nonetheless, activity-based ablation of omission cue-reactive neurons in IL by Daun02 blocked the OCIS of drug seeking, thereby establishing the causality between omission cue-induced neural activity and relapse-suppression.

Taken together, the present results indicate that omission cue-reactive neurons in IL act as a functional cellular unit for relapse-suppression and, therefore, as an anti-relapse neuronal ensemble. This interpretation is consistent with the notion that IL is the brain site responsible for the inhibitory control of not only drug seeking but also learned responses in general[69,70]. However, the existence of neurons reactive to drug-predictive (rather than omission) contexts with the opposite relapse-promoting function has been reported in the same IL cortex[37] (also see refs. [38,71,72]). These seemingly contradictory but neural activity-specific results may explain the mixed reports[33,34] regarding medial prefrontal cortical regulation of drug seeking and relapse, and caution the use of non-activity-specific techniques to probe brain behavioral functions. Electrophysiological evidence that different IL neurons encode response execution and inhibition[73] further confirms the importance of manipulating brain cells based on their intrinsic activity in addition to other characteristics, such as locality and phenotype.

However, such neural activity-based manipulations have also produced seemingly contradictory results. In the current study, Daun02 disruption of IL neurons reactive to cocaine omission cues increased operant response. Paradoxically, in Pfarr et al.[40], Daun02 disruption of IL neurons reactive to alcohol availability cues also increased operant response. These results were each interpreted to attribute a single behavioral function—relapse-suppression—to presumably two distinct units of IL neurons—each reactive to cues signaling drug availability or omission, thereby raising a concern regarding the "cue-specificity" of Daun02 disruption in IL.

The numerous differences in the experimental design between Pfarr et al.[40] and the current study make straightforward comparison of these seemingly contradictory results difficult. Nevertheless, in conjunction with our previous observations[22,23,37], these discrepancies may be explained by differences in the environmental contexts in which Daun02 disruption was achieved. In the current study, Daun02 disruption of IL neurons reactive to cocaine omission cues (which inhibit behavior) was achieved in a well-habituated behaviorally "neutral" context. In Pfarr et al., Daun02 disruption of IL neurons reactive to alcohol availability cues (which excites behavior) was achieved in an "extinguished" alcohol-predictive context (which inhibits behavior)[40]. These arrangements may have simultaneously disrupted two distinct units of IL neurons, each exerting opposing behavioral actions. Consistent with this assumption, Daun02 disruption of IL neurons reactive to an "extinguished" food-predictive context (which inhibits behavior) increased—rather than decreased—operant response, while Daun02 disruption of IL neurons reactive to a "non-extinguished" food-predictive context (which excites behavior) decreased operant response[23]. Similarly, Daun02 disruption of IL neurons reactive to a "non-extinguished" heroin-predictive context (which excites behavior) also decreased operant response[37]. While further research is necessary, it appears important to control all environmental stimuli, including the background context for activity-based brain cell manipulations, to establish the cue-specificity of brain behavioral function.

Using a similar behavioral procedure for non-drug rewards and activity-based neural ablation by Daun02 in a behaviorally "neutral" context, we have demonstrated that two distinct functional units of neurons—each selectively responsible for the opposing behavioral actions of availability and omission cues—co-exist within IL[22]. While the co-existence of distinct neuronal ensembles for relapse-promotion and relapse-suppression within IL and other brain sites still needs to be elucidated, such dissociation in the neurobiological regulation of appetitive behavior may be exploited in the future for the development of medications to facilitate—if not mimic—the anti-relapse action of drug-omission cues. Further studies of omission cue-reactive neuronal ensembles, as well as ensemble-specific brain circuitry and neurochemical processes, may advance current knowledge in addiction medicine through functional characterization of "druggable" targets for relapse prevention.

## Methods

All experimental procedures were conducted in accordance with the National Institutes of Health (USA) Guidelines for the Care and Use of Laboratory Animals and approved by the Institutional Animal Care and Use Committees at University of Maryland Baltimore School of Medicine and The Scripps Research Institute.

**Subjects.** A total of 227 male rats were used. Of these, 177 were Long Evans (LE) and 50 were *Fos-lacZ* transgenic (on Sprague Dawley background)[54]. Long Evans rats were purchased from Charles River, Inc. (Wilmington, MA). *Fos-lacZ* rats were bred at The Scripps Research Institute, and genotyped by Laragen, Inc. (Culver City, CA). Rats weighing 250–300 g at the start of experiments were housed in a temperature and humidity-controlled room, maintained on a 12 h/12 h reverse light/dark cycle. The rats were always trained and tested during the dark (active) phase in dedicated operant conditioning chambers ("chamber"). Each chamber was equipped with two retractable levers (one "active lever" and one "inactive lever"), a "light-cue," a pump, and either a liquid swivel system for cocaine or a drinking well for alcohol. At all times, water and food were available ad libitum.

**Surgery.** The LE rats assigned to the OCIS procedure for cocaine seeking were implanted with an intravenous catheter made of Micro-Renathane (Braintree Science, Braintree, MA) for intravenous cocaine self-administration. All *Fos-lacZ* rats were implanted with the same intravenous catheter as well as permanent bilateral guide cannulae (22G; Plastics One, Roanoke, VA, USA) for the microinjection of Daun02 (4.0 μg/1.0 μl in phosphate-buffered saline containing 5% dimethyl sulfoxide and polysorbated [Tween®] 80) or vehicle into the infralimbic cortex (IL). The microinjection coordinates were anteroposterior + 3.2 mm, mediolateral ± 0.6 mm, and dorsoventral −5.5 mm. Rats were allowed to recover at least 7 days before the start of the behavioral procedures. Daun02 was purchased from Sequoia Research Products, Pangbourne, Berkshire, UK (Cat# SRP0400g).

**Behavioral procedures.** OCIS procedure for cocaine seeking: schematics and timeline are depicted in Fig. 1a. This procedure consisted of three experimental phases. At all times, insertion of active and inactive levers into an operant conditioning chamber signaled the start of a once daily lever-pressing session conducted under a fixed ratio 1 schedule of reinforcement (FR1). A press on the active lever resulted in a single intravenous delivery of cocaine (1.0 mg/kg) or saline. Each delivery of cocaine or saline was paired with 20 s illumination of a light-cue signaling a 20 s time-out period, during which presses on the active lever were recorded but had no scheduled consequence. At all times, presses on the inactive lever were recorded but without a scheduled consequence. Each rat was housed in a dedicated operant conditioning chamber (habituation) to minimize the neurobehavioral impact of environmental stimuli other than those manipulated experimentally.

In summary, a total of 63 rats were initially used to determine the anti-relapse action of cocaine S- (Fig. 1). These rats were randomly assigned to four experimental groups: Compulsivity test, Cocaine (1.0 or 2.0 mg/kg) and Stress priming groups ($n = 14,16,16,17$). Of these, 18 rats were excluded from the study: three rats died during the intravenous surgery, three rats died due to post operation complications, and 12 rats lost the intravenous catheter patency before completing the entire experimental schedule and/or failed to satisfy the preset training criteria (see below). A total of 45 rats were thus retained for the final statistical analyses and graphic representations (see below).

Detailed procedures for each experimental phase are described below:

I. Self-administration training (2–3 weeks): the purpose of this phase was to establish [1] operant responses for cocaine and [2] active lever and light-cue as

"cocaine availability cues". All rats were trained to press an active lever for cocaine presented together with a light cue in once daily 4-h operant conditioning sessions. Rats were required to satisfy the following training criteria: [1] a minimum of 2 weeks of cocaine self-administration and [2] a minimum of 30 cocaine injections for 3 consecutive days. Rats that did not satisfy these criteria within 3 weeks of training were excluded. During this phase, the active lever and light-cue came to predict the availability of cocaine and thereafter served as cocaine-predictive "availability cues" for the remaining experiments.

II. Discrimination training (9–12 weeks): the purpose of this phase was to establish a discriminative stimulus predictive of cocaine omission (S-). The rats were trained to recognize an olfactory cue as S-. Each rat underwent alternating once daily cue-training sessions: [1] "No S- training" (active lever-pressing for cocaine and light-cue in the absence of an orange scent) and [2] "S- training" (active lever-pressing for saline—instead of cocaine—and light-cue in the presence of an orange scent). Each training session lasted for 2 h. The orange scent was provided by placing a Petri dish with gauze soaked with 3.0 ml of orange extract (McCormick® Orange Extract, McCormick & Company, Sparks, MD, USA) inside each chamber, starting 30 min prior to and remaining throughout each session. For No S- sessions, 3.0 ml of water was used instead of the orange extract. Rats were required to satisfy the following criteria: [1] a minimum of 9 weeks of Discrimination training, [2] a minimum of 15 cocaine injections across three consecutive No S- sessions, and [3] a maximum of five saline injections across three consecutive S- sessions. Rats that did not satisfy these criteria within 12 weeks of training were excluded.

III. Discrimination tests (2 days): the purpose of this phase was to test the anti-relapse action of cocaine S-. The rats were randomly assigned to one of three experimental groups defined by the type of priming used: minor electric foot-shock stress or 10 or 20 mg/kg of IP cocaine. Each rat underwent two types of once daily cue-test sessions: [1] "No S- test" (active lever-pressing for saline—instead of cocaine—and light-cue in the absence of an orange scent) and [2] "S- test" (active lever-pressing for saline and light-cue in the presence of an orange scent). The order of S- and No S- tests were randomly counterbalanced between subjects. Each test session lasted for 6 h and was divided into three 2-h blocks designed to determine the anti-relapse action of S- against the relapse-promoting action of cocaine availability cues (first block), stress, or cocaine priming (second and third blocks). As with the S- training sessions, the S- test session was preceded (by 30 min) and accompanied by orange scent. The anti-relapse action of S- against cocaine availability cues (active lever and light-cue) was determined in all rats, while the action against stress and cocaine priming (two different doses) was determined in different groups of rats. Rats in one of the three experimental groups received sham shock (no shock) and then stress priming (10 min of intermittent electric foot-shock) at 2-h and 4-h into each session, respectively (within-session reinstatement). Electric foot-shock (0.8 mA, 1-s) was delivered on a variable time schedule at a mean interval of 40 s (10–70 s range). This stress priming procedure was based on a previously developed protocol[42]. Rats in the remaining two groups received a non-contingent IP injection of saline (0.5 ml) and then one of the two priming doses of cocaine (10 or 20 mg/kg) at 2-h and 4-h into each session, respectively. The orange extract or water was resupplied at 2-h and 4-h into each session. The total number of active-lever presses for each 2-h block was used as the dependent variable for statistical analysis.

Compulsivity test (Fig. 1d): the purpose of this additional experiment was to determine whether the extended access to cocaine through the OCIS procedure resulted in compulsive drug intake. Rats were trained under the OCIS procedure for cocaine (Fig. 1). Half of these rats were tested following Self-Administration training (2–3 weeks of cocaine history), while the remaining rats were tested following Discrimination training (11+ weeks of cocaine history). In the absence of cocaine S-, each rat was allowed two once daily 2-h sessions to self-administer cocaine under a FR1 schedule of reinforcement. Each self-administered cocaine intravenous infusion was paired with a sham shock (0.0 mA) during the first "baseline" session and an electric foot-shock (0.8 mA, 1-s) during the second session (adapted from ref. [18]). The percent reduction in the number of self-administered cocaine infusion (2-h totals) due to the foot-shock was used as a measure of each animal's resistance to punishment (an adverse consequence) or "compulsivity".

OCIS procedure for alcohol seeking: schematics and timeline are depicted in Fig. 2a. This procedure consisted of eight experimental phases. At all times, insertion of both active and inactive levers into an operant conditioning chamber signaled the start of a once daily lever-pressing session under a FR1 schedule of reinforcement. Animals were placed in the operant conditioning chamber for 60 min prior to the start of each session (habituation). A press on the active lever resulted in a single 0.1 ml delivery of either alcohol (20%, v/v) or water into a drinking well. This alcohol dose is known to support robust operant alcohol self-administration without sucrose fading in Long Evans[74] and Wistar[75] rats. Each delivery of alcohol or water was paired with 5 s illumination of the light-cue signaling a 5-s time-out period, during which presses on the active lever were recorded but without a scheduled consequence. At all times, presses on the inactive lever were recorded but without a scheduled consequence. Each rat was maintained on different diets during different experimental phases: standard lab chow (Phases I–III & VII–VIII), alcohol liquid diet adapted from refs. [44,45,47] (Phase IV), and then alcohol liquid diet alternating with alcohol-free liquid diet (Phases V–VI).

In summary, a total of 33 rats were initially used to determine the anti-relapse action of alcohol S- (Fig. 2). Of these, 15 and 18 rats were randomly assigned (purchased) to alcohol and stress priming groups, respectively. Of these, eight rats failed to satisfy the preset training criteria (see below) and were excluded from the study. A total of 25 rats were thus retained for the final statistical analyses and graphic representations.

Before undertaking the OCIS procedure for alcohol (below), all rats were first acclimated to alcohol by receiving concurrent access to two drinking bottles, containing 20% (v/v) alcohol (in water) or plain water, in their home cages for 3 to 4 weeks (adapted from ref. [50]). This was to facilitate operant learning for alcohol. The water bottle was always available, while the alcohol bottle was available intermittently (every other day).

I. Self-administration training (4–5 weeks): the purpose of this phase was to establish [1] operant responses for alcohol and [2] active lever and light-cue as "alcohol availability cues". All rats were trained to press an active lever for alcohol. Each training session lasted for 14 h during the first week and then for 1 h during the remaining weeks. During the first week, the rats underwent three training sessions over 5 days (Mon–Fri); each 14-h session was conducted every other day. Thereafter, the rats were trained 1 h daily. Training criteria: a minimum of 4 weeks of training and a minimum of 20 alcohol deliveries per session for 3 consecutive days. Rats failing to satisfy these criteria within 5 weeks were excluded.

II. Discrimination training NW (8–11 weeks): the purpose of this phase was to establish a discriminative stimulus predictive of alcohol omission (S-) under a non-withdrawal (NW) state. The rats were trained to recognize an olfactory cue as S-. Each rat underwent alternating once daily cue-training sessions: [1] "No S- training" (active lever-pressing for alcohol and light-cue in the absence of an orange scent) and [2] "S- training" (active lever-pressing for water—instead of alcohol—and light-cue in the presence of an orange scent). Each training session lasted for 1 h. The orange scent was provided by placing a Petri dish with gauze soaked with 3.0 ml of an orange extract (McCormick® Orange Extract, McCormick & Company, Sparks, MD, USA) inside each chamber starting 30 min prior to and remaining throughout each session. For No S- training, 3.0 ml of water was used instead of the orange extract. Rats were required to satisfy the following criteria: [1] a minimum of 8 weeks of Discrimination training, [2] a minimum of 20 alcohol deliveries for three consecutive No S- session,s and [3] a maximum of five water deliveries for three consecutive S- sessions. Rats that did not satisfy these criteria within 11 weeks of training were excluded.

III. Discrimination tests NW (2 days): the purpose of this phase was to test the anti-relapse action of alcohol S- under a non-withdrawal state (NW). The rats were randomly assigned to one of two experimental groups defined by the type of priming used: the pharmacological stressor yohimbine or alcohol. Each rat underwent two types of once daily cue-test sessions: [1] "No S- test" (active lever-pressing for water —instead of alcohol—and light-cue in the absence of an orange scent) and [2] "S- test" (active lever-pressing for water and light-cue in the presence of an orange scent). The order of S- and No S- tests were randomly counterbalanced between subjects. Each test session lasted for 3 h and was divided into three 1-h blocks designed to determine the anti-relapse action of S- against the relapse-promoting action of availability cues (first block), as well as stress or alcohol priming (second and third blocks). As with the S- training sessions, the S- test session was preceded (by 30 min) and accompanied by orange scent. The anti-relapse action of S- against availability cues (active lever and light-cue) was determined in all rats, while the action against stress and alcohol priming were determined in different groups of rats. Rats in the stress priming group received non-contingent intraperitoneal injections of saline (0.5 ml) and then yohimbine (0.75 mg/kg, adapted from ref. [51]) at 1-h and 2-h into each session, respectively. Rats in the alcohol priming group received a non-contingent delivery of water (0.2 ml) and then alcohol (0.2 ml, 20%, v/v, adapted from ref. [50]) into the drinking well at 1-h and 2-h into each session, respectively. Prior to the delivery of water or alcohol priming, any remaining fluid (water) in the drinking well was cleared by the experimenter. The orange extract or water was resupplied at 1-h and 2-h into each session. The total number of active-lever presses for each 1-h block was used as dependent variables for statistical analysis.

IV. Dependence induction (3 weeks): the purpose of this phase was to induce alcohol dependence via alcohol liquid diet. All rats were maintained on continuous (24 h; 7 days/weeks) alcohol liquid diet consisting of alcohol (10%, v/v), Boost® nutritional supplement (Nestle USA, Rosslyn, VA, USA), as well as vitamins and minerals (adapted from refs. [44,45]).

V. Discrimination training AW (3–5 weeks): the purpose of this phase was to establish alcohol S- under an acute withdrawal (AW) state. All rats were subjected to the experimental procedures for Discrimination training NW while undergoing acute withdrawal from alcohol. For this, alcohol-free liquid diet (sucrose replacing alcohol to equalize caloric content) was substituted for alcohol liquid diet for 10 h daily. The first operant training session was resumed 4–5 days after this diet schedule change. Each training session was conducted 5–9 h into alcohol withdrawal —a time window associated with physical and affective signs of withdrawal as well as with elevated alcohol intake in dependent rats[44,48,76–78]. Rats were required to satisfy the following criteria: (1) a minimum of 3 weeks of training, (2) a minimum of 20 alcohol deliveries for three consecutive No S- sessions, and (3) a maximum of five water deliveries for three consecutive S- sessions. Rats that did not satisfy these criteria within 5 weeks of training were excluded.

VI. Discrimination tests AW (2 days): the purpose of this phase was to test the anti-relapse action of alcohol S- under an acute withdrawal (AW) state. All rats were subjected to the experimental schedules for Discrimination Tests NW while undergoing acute withdrawal from alcohol. Each testing session was conducted 5–9 h into alcohol withdrawal.

VII. Protracted withdrawal (2 weeks): the purpose of this phase was to induce protracted withdrawal (PW). All rats were maintained on standard lab chow and had no access to alcohol in their home cages.

VIII. Discrimination tests PW (2 days): the purpose of this phase was to test the anti-relapse action of alcohol S- under a protracted withdrawal (PW) state. All rats were subjected to the experimental schedules for Discrimination Tests NW at 2 weeks into alcohol withdrawal —a time period associated anxiety-like behavior[79] and elevated alcohol intake[77].

Localization and phenotypic characterization of omission cue-activated neurons in IL: Rats were randomly assigned to two experimental groups (Fig. 3a, b) and trained under the OCIS procedure to learn an olfactory stimulus (orange scent) as either cocaine S- or alcohol S-. The rats trained for alcohol S- were not subjected to the dependent/post-dependent procedures and thus were both trained and tested under a non-withdrawal state; this was done to isolate neural activity due to omission cues from similar activity due to alcohol withdrawal. Additional rats were habituated to the same olfactory stimulus (orange scent) for 7 days (90 min/day) but were not subjected to the OCIS procedures; these rats never experienced either cocaine or alcohol. All rats were placed in the operant conditioning chamber at least for 60 min prior to the start of each training and testing session (habituation). During Discrimination training, each rat was subjected to alternating once daily sessions to pair "No S- and either cocaine or alcohol" and "S- and either saline or water (i.e., no cocaine or no alcohol)". We were concerned that this routine cycle (the "No S-" training followed by the "S-" training, or vice versa) would result in neural activity due to the "expectancy" or higher basal Fos expression independent of the S- presentation during Discrimination test. The rats thus underwent a "habituation" phase to break the cycle prior to Discrimination test. However, no systematic data is available to determine whether this habituation was indeed necessary. For Discrimination test, the rats were randomly divided into two groups defined by the final cue-condition to induce Fos and exposed to the odor ("S-") or no odor ("No S-") for 90 min (for optimal induction of Fos protein, as well as c-fos mRNA by a "sustained" stimulus, see ref. [80]), then deeply anesthetized with isoflurane and sacrificed to harvest brains for Fos IHC and RNAscope®. For Fos IHC (Fig. 3e), the brains were perfused with 0.1 M phosphate-buffered saline followed by 4% paraformaldehyde in 0.1 M sodium phosphate (pH 7.4), removed, and cut into 40 μm thick coronal sections. For fluorescence in situ hybridization via RNAscope® (Fig. 3f), the brains were "flash frozen" in isopentane, removed, and cut into 10 μm thick coronal sections. For all cases, active lever, light-cue and drug reward (cocaine or alcohol) were withheld to determine neurons specifically reactive to S- (or No S-).

In summary, a total of 81 rats were initially used to determine and characterize omission cue-activated neurons (Fig. 3): 37 rats for cocaine S-, 32 rats for alcohol S-, and 12 rats for the olfactory control. Of the rats assigned for cocaine S-, two rats died during the intravenous surgery, one rat died due to post surgery complications, seven rats failed to satisfy the preset training criteria or lost IV catheter patency, and one rat (exposed to cocaine S- for RNAscope®) was excluded as an outlier with unusually high c-fos expression (> 2 standard deviations of the mean). Of the rats assigned for alcohol S-, seven rats failed to satisfy the preset training criteria. A total of 63 rats were retained for the final statistical analyses and graphic representations (see below): 26 rats for cocaine S- (19 rats for Fos IHC and 7 rats for RNAscope®), 25 rats for alcohol S- (17 rats for Fos IHC and 8 rats for RNAscope®), and 12 rats for the olfactory control (all for Fos IHC).

Functional characterization of omission cue-activated neurons in IL: Fos-lacZ transgenic rats were trained under the OCIS procedure to establish an olfactory stimulus (orange scent) as cocaine S- (Fig. 4a). Animals were placed in the operant conditioning chamber at least for 60 min prior to the start of each training and testing session (habituation). The rats were then randomly divided into four groups (Group) defined by cue ("S-" or "No S-") and microinjection ("Daun02" or "vehicle") for neural activity-targeted disruption. For this, each rat was exposed to either S- or No S- for 90 min, then received a bilateral microinjection of Daun02 (2.0 μg/0.5 μl/side) or vehicle (0.5 μl/side) into IL (Fig. 4b). Active lever, light-cue, and cocaine were all withheld to target neurons specifically reactive to S- (or No S-). Each rat was allowed to recover for 2 days in their home cages and then subjected to Discrimination tests for cocaine S-, as described above. Daun02 disruption was verified by Fos IHC. For this, all rats were exposed to S- for 90 min, deeply anesthetized with isoflurane, and then perfused with 0.1 M phosphate-buffered saline followed by 4% paraformaldehyde in 0.1 M sodium phosphate (pH 7.4). Active lever, light-cue and cocaine were all withheld to determine the neurons specifically reactive to S- (or No S-). The brains were removed, cut into 40 μm thick coronal sections, and processed for Fos IHC (Fig. 4e).

In summary, a total of 50 rats were used to determine the anti-relapse neurons in IL (Fig. 4). Of these, 16 rats were excluded from the study: four rats died during either intravenous or intracranial surgery, three rats died due to post operation complications, and nine rats failed to satisfy the preset training criteria or lost IV catheter patency. An additional four rats were excluded based on the histological verification of the intracranial cannulae (see below). A total of 30 rats were thus retained for the final statistical analyses and graphic representations (see below).

**Fos and β-galactosidase immunohistochemistry.** The paraformaldehyde-fixed brains were sectioned and processed for Fos immunohistochemistry (IHC) and quantified. For this, Fos antibody (1:2,000 dilution) from Cell Signaling Technology (Cat# 2250S, Danvers, MA, USA; RRID: AB_2247211) was used. These sections were developed using an ImmPRESS™ HRP (Peroxidase) Polymer Kit from Vector Laboratories (Cat# MP-7451, Burlingame, CA, USA; RRID:AB_2631198) and diaminobenzidine. β-galactosidase antibody (1:1000 dilution) from Santa Cruz Biotechnology (Cat# sc-65670, Dallas, TX, USA; RRID:AB_831022). Additional brain sections were processed for X-gal (5-bromo-4-chloro-3-indolyl β-D-galacto-pyranoside) histochemistry to validate the presence of β-galactosidase. The X-gal kit (Cat# XGAL-0100; RRID:AB_2631199) was purchased from Rockland Immunochemicals Inc. (Pottstown, PA).

The histological procedures for Fos IHC were based on previously published procedures[22,37,40,41]. For this, bright-field images of IL were captured and digitized using an EVOS microscope (ThermoFisher, Inc., Waltham, MA, USA). These images were used for [1] histological verification of the co-localization of Fos and β-galactosidase (i.e., Fos-lacZ positive), [2] histological verification of the microinjection sites, and [3] histological quantification of Fos-expressing nuclei. Nuclei expressing Fos and/or β-galactosidase were counted using ImageJ (National Institute of Health, Bethesda, MD, USA; RRID:SCR_003070). The threshold level was set to detect moderately to darkly stained nuclei but not lightly stained nuclei. We counted nuclei from sampling areas around the microinjection site from 3–5 coronal sections per rat. Average numbers of Fos-positive (Fos+) nuclei per mm² calculated for each rat were used for statistical analyses and data representations. Image capture and quantification were conducted by an observer blind to the experimental conditions. Two rats were excluded because their microinjection sites were determined to be outside of IL. Two additional rats were excluded due to necrosis in tissues surrounding the microinjection sites.

**4-plex fluorescence in situ hybridization via RNAscope®.** The flash frozen brains were cut into 10 μm coronal sections, placed directly onto charged microscope slides, stored at −20 ℃, and processed in accordance with the user manuals for *RNAscope® Multiplex Fluorescent Reagent Kit v2* (Advanced Cell Diagnostics, #323100-USM) and *RNAscope® 4-plex Ancillary Kit for Multiplex Fluorescent Reagent Kit v2 Technical Note* (Advanced Cell Diagnostics, #323120-TN). Briefly, frozen sections were fixed in neutral buffered formalin, dehydrated with ethanol, and pretreated with hydrogen peroxidase and then protease digestion. Four different RNA-specific probes were applied (see below) and hybridized. To identify and quantify different mRNAs, sections then underwent three separate hybridization-amplification steps and, finally, four separate signal development steps for each unique fluorophore–RNA complex.

On each brain section, we targeted four mRNA types (encoding protein): *c-fos* (Fos), *Slc17a7* (vesicular glutamate transporter 1 [VGLUT1]), *Slc32a1* (vesicular gamma-aminobutyric acid transporter [VGAT]), and *CHAT* (choline acetyltransferase [ChAT])—each as the marker for omission cue-activated (S-reactive), glutamatergic (GLU), GABAergic (GABA) and cholinergic (ACh) cells, respectively—as well as DAPI—as the marker for DNA-expressing nuclei. All mRNA probes were designed by Advanced Cell Diagnostics: *c-fos* (GenBank accession # NM_022197.2: target region, 473–1497), *Slc17a7* (GenBank accession # NM_053859.2: target region, 529–11630), *Slc32a1* (GenBank accession # NM_031782.21: target region, 288–1666), and *CHAT* (GenBank accession #NM_001170593.1: target region, 259–1141). Each probe was tagged with a unique Opal™ fluorophore from PerkinElmer, Inc. (Waltham, MA, USA): *c-fos*-Opal 690 (Cy5), *Slc17a7*-Opal 520 (FITC), *Slc32a1*-Opal 570 (Cy3), *CHAT*-Opal 620 (Texas Red), and DAPI.

Fluorescent images of the brain sections for RNAscope® were captured using ZEISS LSM 710 and 780 laser scanning confocal microscopes with ZEN image software (Carl Zeiss, Oberkochen, Germany). These systems are capable of continuous spectral detection with seven lasers (405, 458, 488, 514, 561, 594, 633) and a fully tunable (within 10 nm intervals) Quasar multi-channel system. Images were saved as 64-bit TIFF files and analyzed by Image-Pro Premier software (Media Cybernetics, Rockville, MD, USA). These images were used to determine the percentages of [1] *c-fos* + (S- reactive), *Slc17a7* + (GLU), *Slc32a1* + (GABA), and *CHAT* + (ACh) phenotypes within DAPI + nuclei (Fig. 3h), [2] *c-fos* + (S-reactive) phenotypes within *Slc17a7* + (GLU), *Slc32a1* + (GABA), and *CHAT* + (ACh) nuclei (Fig. 3i), and [3] *Slc17a7* + (GLU), *Slc32a1* + (GABA), and *CHAT* + (ACh) phenotypes in *c-fos* + (S- reactive) nuclei—all localized in IL (Fig. 3d).

To identify each mRNA signal, we used brain sections taken from the same animals, but prepared without any probe as the negative control, to set the threshold for background signals (noise), such as the natural auto-fluorescence activity. We used DAPI+ nuclei within IL as the "region of interests" (ROIs) to identify and quantify each neural phenotype. On average (± SEM), we analyzed 1946.2 ( ± 200.1) ROIs (DAPI+ nuclei) per animal. Within each ROI (single DAPI + nucleus), we used correlation coefficients (ratios between [1] co-localized area of DAPI and a specific gene target and [2] the total area of each DAPI-expressing nucleus) as the basis to identify each phenotype. Specifically, we applied a 25% or greater correlation coefficients between (1) DAPI and *c-fos*, (2) DAPI and *Slc17a7*, (3) DAPI and *Slc32a1*, or (4) DAPI and *CHAT* signals as the criteria for (1) S-reactive, (2) GLU, (3) GABA, and (4) ACh phenotypes. These criteria were used to minimize "false positives" due to background (e.g., auto-fluorescence), non-

specificity (e.g., unspecific bindings due to staining or manufacture mistakes), or imaging limitation (e.g., it is not possible to separate *Slc17a7* signals on a GLU cell body from *Slc17a7* signals on GLU pre-synaptic terminals without using an electron microscope). We also took account of the fact that *c-fos* mRNA can be transiently expressed by any neuron (or even non-neural astrocytes) in response to any stimulus—external (e.g., cage change) or internal (e.g., circadian surge in metabolic signals, such as leptin)—other than an experimentally-manipulated stimulus (e.g., S-) at any given time. We finally validated these criteria visually on each brain section. For example, we confirmed that the average correlation coefficient (± SEM) between DAPI and *c-fos* for all remaining DAPI+ nuclei, which did not satisfy the preset criterion (> 25%) for the "S- reactive" phenotype, was 0.09% (± 0.01%).

For data analysis (see below), total numbers of nuclei per mm$^2$ that satisfied each phenotypic criterion were used. For the graphic representations to depict the phenotypic composites of omission cue-activated neurons in IL, percentages (%) of each phenotype within a specific "parent" phenotype were used: % of *c-fos+* /*Slc32a1+*/*Slc32a1+*/*CHAT+* /*unclassified* (RNAscope signal negative) nuclei within all DAPI+ nuclei (Fig. 3h: % of S- reactive/GLU/GABA/Ach/other cells within all cells), % of *c-fos+* nuclei within *Slc32a1+*/*Slc32a1+*/*CHAT+*/*unclassified* nuclei (Fig. 3i: % S- reactive cells within GLU/GABA/Ach/other cells), and % of *Slc32a1+*/*Slc32a1+*/*CHAT+*/*unclassified* nuclei within *c-fos+* nuclei (% of GLU/GABA/Ach/other cells within S- reactive cells).

**Statistical analysis**. For all cases, parametric statistical analyses were used. When appropriate, analysis of variances (ANOVAs) were followed by post-hoc Tukey honestly significant difference (HSD) or Bonferroni test. For all cases, differences were considered significant when $P < 0.05$ (two-tailed). As our multifactorial ANOVA yielded multiple main and interaction effects, we only report significant effects that are critical for data interpretation. No statistical methods were used to predetermine sample sizes, but our sample sizes were similar to those reported in the relevant literature. Data distribution was assumed to be normal, but this was not formally tested. We used SigmaPlot/SigmaStat version 12.5 (Systat Software, San Jose, CA, USA) and IBM SPSS Statistics 23 and 25 (IBM Corporation, Armok, NY, USA).

OCIS procedure for cocaine: the behavioral procedures for the initial 2-h block of each 6-h session of Discrimination test were identical across the three experimental groups (stress, 10 or 20 mg/kg cocaine priming groups) designated to establish the anti-relapse action of cocaine S- (Fig. 1e). Thus, the initial 2-h totals of active lever-presses were pooled together and analyzed by Student's *t*-test for repeated measures (paired) with discrimination cue or "Cue-test" (two levels: S- vs. No S-) as within-subjects factor. The second and third 2-h totals of active-lever presses (within-session reinstatement) were analyzed within each group by two-way ANOVA for repeated measures with Cue-test (two levels) and relapse-promoting stimulus or "Priming" (two levels: no priming [no shock or saline] or priming [mild foot-shock, 10 or 20 mg/kg of cocaine]) as within-subjects factors (Fig. 1f, g, h). The results of Compulsivity tests were analyzed using Student's *t*-test (non-paired) with cocaine self-administration history (two levels: 2+ weeks and 12+ weeks) as between-subjects factor (Fig. 1d).

OCIS procedure for alcohol: the total active lever-presses during Discrimination training NW (last 5 "No S-" sessions) and Discrimination training AW (first 5 "No S-" sessions) were analyzed together by two-way ANOVA for repeated measures with withdrawal state or "Withdrawal" (two levels) and training session or "Session" (five levels) as within-subjects factors (Fig. 2c, d). The behavioral procedures for the initial 1-h block of each 3-h Discrimination test were identical for the two experimental groups (stress and alcohol priming) designated to establish the anti-relapse action of alcohol S- (Fig. 2e). Thus, the initial 1-h totals of active lever-presses were pooled together and analyzed by two-way ANOVA for repeated measures with alcohol withdrawal state or "Withdrawal" (three levels: non-withdrawal [NW], acute withdrawal [AW] and protracted withdrawal [PW]) and discrimination cue or "Cue-test" (two levels: S- and No S-) as within-subjects factors. The second and third 1-h totals of active-lever presses (within-session reinstatement) were analyzed within each group by three-way ANOVA for repeated measures with Withdrawal (three levels), Cue-test (two levels) and relapse-promoting stimulus or "Priming" (two levels: no priming [saline or water] or priming [yohimbine or alcohol]) as within-subjects factors.

Localization and phenotypic characterization of omission cue-activated neurons in IL: the Fos IHC results (Fos+ nuclei per mm$^2$) to determine omission cue-induced neural activation in IL (Fig. 3g) were analyzed by two-way ANOVA with training group or "Group" (three levels: cocaine S-, alcohol S- and olfactory control [Fig. 3a–c]) and cue-test conditions or "Cue-Test" (two levels: S-/odor cue vs. No S-/No odor cue) as between-subjects factors. Initial observations under the microscope revealed that layer I contained little to no Fos+ nuclei. This observation was systematically confirmed by additional analyses to determine S- reactive nuclei in different cortical layers via RNAscope® (see below). The subsequent Fos IHC as well as RNAscope® analyses were thus focused on layers II, III, and V/VI.

RNAscope® results (*c-fos+*/*Slc17a7+*/*Slc32a1+*/*CHAT+*/unclassified nuclei per mm$^2$), for determining S- reactive, GLU, GABA, ACh, and "other" phenotypes from rats trained for cocaine S- vs. alcohol S- (Fig. 3h), were analyzed by two-way mixed ANOVA with training group or "Group" (two levels: cocaine S- and alcohol S-) as between-subjects factor and cellular phenotype or "Phenotype" (five levels) as within-subjects factor. RNAscope® results (*c-fos+* nuclei within *Slc17a7+* /*Slc32a1+*/*CHAT+*/unclassified nuclei per mm$^2$), for determining S- reactive nuclei within GLU, GABA, ACh, and other phenotypes from rats trained for cocaine S- vs. alcohol S- (Fig. 3i), were analyzed by two-way mixed ANOVA with Group (two levels) as between-subjects factor and Phenotype (four levels) as within-subjects factor. RNAscope® results (*Slc17a7+*/*Slc32a1+*/*CHAT+* /unclassified nuclei within *c-fos+* nuclei per mm$^2$), for determining GLU, GABA, ACh, and other phenotypes within S- reactive nuclei from rats trained for cocaine S- vs. alcohol S- (Fig. 3j), were also analyzed by two-way mixed ANOVA with Group (two levels) as between-subjects factor and Phenotype (four levels) as within-subjects factor.

Additional RNAscope® analyses were conducted to determine the expression of cocaine S- or alcohol S- reactive IL nuclei across different cortical layers (Supplementary Fig. 3). For this, the numbers of *c-fos+* nuclei per mm$^2$ within layers I, II, III, and V/VI were determined. These results were analyzed by two-way mixed ANOVA with Group (two levels) as between-subjects factor and cortical layers (four levels: layers V and VI were jointly analyzed) as within-subject factor. Please note: these cortical layer analyses were conducted using additionally processed sections from the same subjects included in the results depicted in Fig. 3 (seven rats for cocaine S- and eight rats for alcohol S-). This resulted in slight variations in S- reactivity (the percentages of *c-fos+* nuclei within DAPI+ nuclei) within the same Group (cocaine S- or alcohol S-) between the results depicted in Fig. 3 and Supplementary Fig. 3.

Functional characterization of omission cue-activated neurons in IL: the initial 2-h totals of active lever-presses were analyzed by two-way mixed ANOVA with experimental group or "Group" (four levels: "No S- & vehicle", "No S- & Daun02", "S- & vehicle", "S- & Daun02") as between-group factor and discrimination cue or "Cue-test" (two levels: "S-" vs. "No S-") as within-subjects factor (Fig. 4c). The second and third 2-h totals of active-lever presses (within-session reinstatement) were analyzed separately for each Group (four separate analyses) by two-way ANOVA for repeated measures with Cue-test (two levels) and Priming (two levels) as within-subject factors. The Fos IHC results, for validating the Daun02 disruption of S- reactive neurons, were analyzed by one-way ANOVA with Group (four levels) as between-group factor (Fig. 4e).

## Data availability

The data that support the findings of this study are available from the corresponding authors upon reasonable request.

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

## Acknowledgements

This work was supported by the Extramural and Intramural funding from National Institute on Drug Abuse as well as National Institute of Alcohol Abuse and Alcoholism, National Institute of Health, USA: R21DA033533 (N.S.), R01DA037294 (N.S.), R01AA023183 (N.S.), R01AA021549 (F.W.), ZIADA000467 (B.T.H.), N01DA59909 (G.I. E.). A.L. and H.N. were supported by Ruth L. Kirschstein Institutional National Research Service Award from National Institute of Alcohol Abuse and Alcoholism, National Institute of Health, USA: T32AA007456 (PIs, Drs. Loren "Larry" Parsons and Michael Taffe). We thank Drs. Jennifer M. Bossert (NIDA/NIH/IRP) and Andree Lessard (University of Maryland, Baltimore) for technical assistance. We thank Drs. Thomas Jhou (Medical University of South Carolina) and Stephan Steidl (Loyola University) for the insightful comments which helped to polish this manuscript. We also thank Dr. Roy A. Wise (NIDA/IRP) for his intellectual support and inspiration throughout this project. Finally, we thank Dr. Larry Parsons (1964-2016) for his assistance in designing the alcohol experiments and warm encouragement. This is publication number 29812 from The Scripps Research Institute.

## Author contributions

N.S., G.I.E., and B.T.H. conceived the research. N.S., G.I.E., B.T.H., F.W., and A.L. designed the research. N.S., G.I.E., and F.W. developed the behavioral procedures. B.T.H. and E.K. developed the molecular procedures. A.L., H.N., and N.S. developed the histological procedures. N.S., A.L., G.L.D., G.E.W., A.C., and T.K. performed the behavioral procedures. A.L., H.N., D.W., G.L.D., and G.E.W. performed the molecular and histological procedures. N.S., G.I.E., and A.L. analyzed the behavioral results. N.S., A.L., H.N., and E.K. analyzed the molecular and histological results. N.S., A.L., G.I.E., F.W., B.T.H., and E.K. wrote the manuscript. N.S. coordinated this work. All authors helped with data interpretation and manuscript editing.

## Additional information

**Competing interests:** The authors declare no competing interests.

