## [Peer Review File · Nature Communications]

Reviewers' comments:

Reviewer #1 (Remarks to the Author):

Preventing relapse is important for the treatment of any kind of addictions. In this study, authors showed the anti-relapse action of environmental cues that signal drug omission in rats using Pavlovian conditioning based model. The omission cues suppressed the behaviors in Cocaine as well as Alcohol case. The omission cues activated c-fos expression in the infra limbic cortex. Authors examined the role of omission cues activated cells on the behavior. These line of experiments are well conducted and the manuscript is well-written. However, to make the manuscript stronger, I have a couple of suggestion.

1. In Fig.4 experiments, authors conducted 2 groups, No S- and S-. However, Authors should examine another control group; S- phase II and No S- in phase III. This experiment tells the specificity of the omission-related cells in the IL.

2. In Fig. 3, It might be very informative to identify the distribution of c-fos positive cells in the IL, since each layer of cells have different projection patterns and if authors see the difference, it is better to discuss the possible relationship with other brain areas.

3. While omission related engrams that author found is new, since neocortical engram cells have been already reported couple of time, it should be better to cite other neocortical engrams like (Ye et al, Cell. 2016) and Kitamura et al., Science, 2017).

Reviewer #2 (Remarks to the Author):

I enjoyed reading this manuscript, and find it of potential broad interest, as well as an advance in our understanding of how alcohol and cocaine seeking are regulated.

Major concerns/questions/suggestions:

1) The premise of the paper is that the OCIS is an innovative translational procedure to increase craving suppression and thus indirectly decrease relapse. However, one could argue OCIS follows the same basic idea as extinction training. In the extinction context the "new rule" of drug unavailability drives active refraining and decreased lever pressing. Moreover, ABA renewal procedures (originally proposed by Bouton & Bolles using conditioned fear in 1979 but widely used to induce reinstatement to drugs since) showed that it was possible to prevent lever pressing by changing the context (the equivalent of introducing the orange extract smell in OCIS). Inhibition of drug seeking through extinction is well described in the literature, as acknowledged by the authors in the introduction (refs 10-14), but the authors only skim over the ABA procedure in the discussion (refs 45,46) and should detail more clearly how the OCIS proposed here differs from the ABA procedures.

2) The sensory modalities used for availability/unavailability are very different (vision for light cue vs smell for the orange sent cue). Authors should discuss how this could impact their findings.

3) To study the ensemble in the IL, rats are sacrificed 90 min after exposure to the orange extract sent. As pointed out by the authors, this time allows optimal induction of c-fos mRNA. However, previous literature shows that rapid (within 15min) transient changes occur in the nucleus accumbens during reinstatement and extinction. Did the authors ever looked at an earlier time-point?

4) The inactive lever presses are only reported during the first 10-14 SA sessions (Figures 1B and 2B). It would be interesting to disclose (as supplementary information if preferred) the inactive lever presses during the different post-SA stages, notably discrimination and the three types of reinstatement.

5) In the first part of the study, rats undergo prolonged access to cocaine (2-3 weeks, 4h sessions) in order to increase compulsive intake despite adverse consequences. Did the authors

ever conduct the OCIS procedure in standard short access cocaine SA? High levels of cue-, prime- and stress-induced reinstatement are observed after short access, would S- also prevent relapse then?

6) The cocaine-paired cue lasts for 20s, in contrast of the alcohol-paired one that lasts 5s. Is there any reason behind this large difference?

7) Daun02 disruption induced a 60% decrease in Fos+ cells (Figure 4E). Did the authors perform an RNAscope experiment to determine the neural phenotype of the 40% remaining Fos+ cells? OCIS is prevented despite the survival of this 40% of cells, do the authors hypothesize the remaining cells display the same phenotype (and this cell type can be discarded for OCIS) or do these cells form a heterogeneous pool of neurons randomly activated during the exposure to the orange extract sent, ie how do the authors explain that "surplus" of Fos+ cells?

Minor concerns/questions/suggestions:

1) Mention the species studied in the title.

2) In line with NIH guidelines to consider sex as a biological variable, mention earlier in the manuscript (in the introduction?) that the study was performed exclusively in males. Consider performing the key experiments (response to cocaine or alcohol availability cues for No S- and S- groups, cue-reinstatement after Daun02 disruption) in a cohort of female rats and compare to males' response.

3) Consider using the terms "cues-/drug/stress-induced reinstatement" throughout the text and figures. It is more widely found in the literature and less confusing than terms like "relapse-promotion", "discrimination tests" or "within-session extinction".

4) In the figures, consider removing "Saline SA" when presenting the discrimination tests/reinstatements. It might be understood as the data presented represents the behaviour of a control group. Stating that animals receive saline instead of cocaine infusions during reinstatements in the methods is clear enough.

5) It is possible that rats experience distress in presence of the orange extract sent since it is linked to withdrawal. Did the authors perform any stress/anxiety test on trained rats in presence of the sent?

6) In the alcohol part of the study, active lever pressing induced alcohol/water delivery in a well. Did the authors control for actual intake of the liquid?

Reviewer #3 (Remarks to the Author):

In this manuscript Laque et al. report the results of a series of studies on the ability of cues signaling drug unavailability to suppress "relapse" to drug and alcohol seeking, and the involvement of infralimbic neurons activated by these cues in suppressing drug seeking behaviors. Overall the studies are labor intensive and represent a valuable contribution to the field. The statistical analysis is robust and appropriate, and the studies appear well-conducted and well controlled, for the most part. I do have some concerns and questions about specific aspects of the study design and control groups, that slightly attenuate my enthusiasm, described below:

Major Points:

1. The evidence presented for the importance of this specific S- activated IL ensemble in suppressing cocaine seeking is somewhat weak. Given previous work from some of the authors (Pfarr et al 2015) showing that ablation of IL neurons activated by alcohol cues results in elevated alcohol seeking behavior, how specific is this IL neural ensemble for suppression of drug and alcohol seeking? Here, the authors only look at the effects of ablated neurons activated during S- or no S-. What if the authors had ablated IL neurons activated by the cue light predicting drug delivery – might those neurons also be important for suppression of drug seeking? This would be a stronger control.

2. In the alcohol experiments all rats appear to receive liquid alcohol diet at the same time; changes in self-administration and relapse during the later tests are then attributed to alcohol

dependence and withdrawal. But the effects of additional self-administration training with alcohol cannot be ruled out as there does not appear to be a parallel control group that receives additional training without the intervening alcohol diet. Additionally, the authors don't demonstrate any dependence or withdrawal specific behavioral effects that can be linked to alcohol seeking, other than a modest increase in alcohol self-administration after the liquid diet exposure period. Given that S- cues have the same effect before and after the alcohol dependence induction, this does not seem to be a critical aspect of the study. But the authors should either modify their text to include this caveat, or include other further clarification.

3. Please clarify whether the S- and no S- discrimination tests were counterbalanced for order.

4. Describing the initial cue availability tests as "relapse tests" seems inaccurate, as rats are not increasing responding from a pre-test baseline (lowered based on extinction or punishment or some other manipulation), but are maintaining responding from the previous discrimination based on learning. For the no S- sessions these appear to be identical to the end of discrimination training.

Minor Points

1. In the introduction the authors make a point of saying that their conditions are more "translationally-relevant" – could they be more explicit about the ways in which these studies are more translationally relevant than previous work?

2. The description of the relapse tests in the Results section is somewhat confusing. The Methods section clarifies that each test consisted of a cue availability period, followed by saline or no shock, followed by cocaine or alcohol or shock, but this basic structure should be more clear earlier in the manuscript.

3. How frequent were the ethanol training sessions? The methods section states that the first week of training ("5 sessions") consisted of 14 hr training sessions, but Figure 2B only has 3 sessions at 14 hours. Were the rats trained with alcohol every other day, or were only 3/5 sessions that week 14 hrs long?

4. There is an error in the Figure 2F/G legend: only 2 conditions are listed, each twice

5. What is the purpose of the post-training habituation phase (1 wk) in Figure 3; I could not find any mention of this in the text

Reviewers' comments:

Reviewer #1 (Remarks to the Author):

Preventing relapse is important for the treatment of any kind of addictions. In this study, authors showed the anti-relapse action of environmental cues that signal drug omission in rats using Pavlovian conditioning based model. The omission cues suppressed the behaviors in Cocaine as well as Alcohol case. The omission cues activated c-fos expression in the infra limbic cortex. Authors examined the role of omission cues activated cells on the behavior. These line of experiments are well conducted and the manuscript is well-written. However, to make the manuscript stronger, I have a couple of suggestion.

1. In Fig.4 experiments, authors conducted 2 groups, No S- and S-. However, Authors should examine another control group; S- phase II and No S- in phase III. This experiment tells the specificity of the omission-related cells in the IL.

We agree that this is an important issue regarding the “neural activity-specificity” of Daun02 effects. This question asks whether the microinjection of Daun02 into IL – independent of the S- cue to induce local neural activity – would have been sufficient to prevent the anti-relapse action of S-. It asks, in other words, what the effects would be in a group of rats that was (1) trained to recognize S- during Phase II but (2) given Daun02 without exposure to S- (i.e., “No S-”) during Phase III – a condition that would have resulted in minimal/basal neural activity in IL (see Fig 3G – “No S-” conditions).

In short, the suggested control groups were already included in our study to establish the neural activity-specificity of Daun02 effects. However, the experimental procedures were perhaps not adequately described. We have thus revised the texts, and also describe the experimental details below:

During Phase II (Discrimination training to learn an odor cue as S-), all rats were subjected to both S- and No S- training. The aim here was to train the rats to learn the association between S- and drug *un*availability (omission) and to develop a neuronal representation of this association.

Prior to Phase III (Daun02 disruption of S- reactive neurons in IL), the rats were randomly subdivided into 4 groups, defined by disruption-cue type (2 levels: “S-” or “No S-”) and microinjection type (2 levels: “Daun02” or “vehicle”) for neural activity-targeted disruption of “S- reactive” neurons in IL. The 4 groups (depicted in Figs 4C/D) are the following:

- Group 1 (“No S- & vehicle”)
- Group 2 (“No S- & Daun02”)
- Group 3 (“S- & vehicle”)
- Group 4 (“S- & Daun02”)

Of these, both Group 1 and Group 2, which were trained with S- during Phase II but exposed to No S- during Phase III, serve as the requested control group (“S- phase II and No S- in phase III”).

During Phase III, each rat was first exposed to either S- or No S- for 90 min and then received a bilateral microinjection of either Daun02 or vehicle into IL (Fig 4B). Active lever, light-cue and cocaine were all withheld to target neurons specifically reactive to S- (or No S-). In *Fos-lacZ* rats (used in the Fig 4 experiment), Daun02 (inactive compound) is catalyzed into daunorubicin (cytotoxin) by beta-galactosidase (enzyme) only in Fos+ “activated” (S- reactive) cells, thereby triggering apoptosis. This effect was expected in Group 4 (S- & Daun02) but not in other groups: Group 1 (No S- & vehicle), Group 2 (No S- & Daun02) and Group 3 (S- & vehicle).

After recovery, each rat underwent Phase IV (Discrimination tests to determine the anti-relapse action of S-: Figs 4C-D). In Group 1 (No S- & vehicle) and Group 3 (S- & vehicle), which received vehicle – rather than Daun02 – during Phase III, the anti-relapse action of S- was preserved. In Group 2 (No S- & Daun02), which received Daun02 without being exposed to S- (i.e., No S-) during Phase III, the anti-relapse action of S- was also preserved. In Group 4 (S- & Daun02), Daun02 disruption of S- reactive neurons in IL blocked the anti-relapse action of S-. The subsequent histological analysis (Fig 4E) indicates that the number of S- reactive neurons was significantly lower only in Group 4 (S- & Daun02), thus validating the Daun02 disruption.

Taken together, Daun02 in IL – without exposing the S- trained rats to S- (i.e., “S- phase II and No S- in phase III”) – did not produce a significant effect on the anti-relapse action of S-, thereby establishing the “neural activity-specificity” of Daun02 effects.

2. In Fig. 3, It might be very informative to identify the distribution of c-fos positive cells in the IL, since each layer of cells have different projection patterns and if authors see the difference, it is better to discuss the possible relationship with other brain areas.

We have conducted additional RNAscope analyses (Supplemental Fig 3) to determine the distribution of c-fos positive (S- reactive) nuclei across different cortical layers: layers I, II, III and V/VI (based on Perez-Cruz et al. 2007, *Neural Plast*, 2007:46276). While no significant difference was found among layers II-VI, S- reactive nuclei were minimal and significantly fewer in layer I. We have described these results and have also discussed “the possible relationship with other brain areas”.

Please note 1: these cortical layer analyses were conducted using additionally processed sections from the same subjects included in the results depicted in Fig 3 (7 rats for cocaine S- and 8 rats for alcohol S-). This resulted in slight variations in reported S- reactivity (the percentages of *c-fos+* nuclei within DAPI+ nuclei) within the same Group (cocaine S- or alcohol S-) between the results depicted in Fig 3 and Supplemental Fig 3.

Please note 2: During the initial RNAscope analysis (described in the original submission of this paper), we already noticed that the expression of c-fos was minimal in layer I, which is known to contain much fewer neurons than other cortical layers. We thus excluded this most superficial layer from the subsequent analyses, as depicted in Figs 3H/I/J, in the original submission. We have specified these facts in this revision.

3. While omission related engrams that author found is new, since neocortical engram cells have been already reported couple of time, it should be better to cite other neocortical engrams like (Ye et al, Cell. 2016) and Kitamura et al., Science, 2017).

We have cited these studies in this revision.

Reviewer #2 (Remarks to the Author):

I enjoyed reading this manuscript, and find it of potential broad interest, as well as an advance in our understanding of how alcohol and cocaine seeking are regulated.

Major concerns/questions/suggestions:

1) The premise of the paper is that the OCIS is an innovative translational procedure to increase craving suppression and thus indirectly decrease relapse. However, one could argue OCIS follows the same basic idea as extinction training. In the extinction context the “new rule” of drug unavailability drives active refraining and decreased lever pressing. Moreover, ABA renewal procedures (originally proposed by Bouton & Bolles using conditioned fear in 1979 but widely used to induce reinstatement to drugs since) showed that it was possible to prevent lever pressing by changing the context (the equivalent of introducing the orange extract smell in OCIS). Inhibition of drug seeking through extinction is well described in the literature, as acknowledged by the authors in the introduction (refs 10-14), but the authors only skim over the ABA procedure in the discussion (refs 45,46) and should detail more clearly how the OCIS proposed here differs from the ABA procedures.

These are all important comments regarding the learning processes underlying the suppression of operant drug response under the OCIS procedure. We agree that the OCIS procedure shares similarities with “extinction”; in particular, the procedure under the “ABA” renewal model for operant drug response (reinstatement). This “extinction” procedure under the ABA renewal model is distinct from the “extinction” procedure under the more traditional “AAA” model of drug reinstatement.

As noted, we already discussed the similarities and differences between the extinction procedure under the AAA model and the OCIS procedure in the original text. For this revision, we have discussed the similarities and differences between the extinction procedure under the ABA renewal model and the OCIS procedure described in the text. Below we provide additional details regarding this matter:

In the ABA renewal model, animals undergo three phases: (1) operant conditioning in Context A, (2) extinction of operant response (established in Context A) in Context B and (3) renewal or reinstatement of operant response (extinguished in Context B) in Context A. Contexts A and B differ in their auditory (e.g., fan turned on vs. off), visual (e.g., red vs. white house light), tactile (e.g., rods of different sizes in floor), circadian (e.g., morning vs. afternoon) and/or olfactory (e.g., orange vs. almond) cues.

Similar to our study, context B signals drug *unavailability* and thus arguably acts as a “drug omission cue”. This “extinction context” is effective against the relapse-promoting action of drug availability cues. However, to our knowledge, the anti-relapse action of Context B against the relapse-promoting action of stress and drug priming has not been determined yet.

Notably, there are several procedural differences between the ABA renewal and the OCIS procedures, such as (1) extensive contrasting between drug availability and drug omission conditions through omission cue/context training (ABA = No; OCIS = Yes) and (2) control for the ambivalent “background” stimuli (e.g., operant conditioning chamber) through extensive habituation training (ABA = No; OCIS = Yes). Future studies need to determine whether such procedural differences affect both anti-relapse action and neuronal representation of drug omission context/cues established under the ABA renewal model and the OCIS procedure.

Finally, “extinction” under the ABA renewal model is different from the “extinction” under the AAA model, which is established in the context where operant response was first established (Context A). Critically, this procedural difference is relevant to understanding the seemingly contradictory results between the current study and Pfarr et al. (2015, *J Neurosci*, 35: 10750-61; see response to Comment 1 from Reviewer 3 below).

2) The sensory modalities used for availability/unavailability are very different (vision for light cue vs smell for the orange scent cue). Authors should discuss how this could impact their findings.

This is an important comment concerning whether the behavioral (response suppression) and neuronal (activation in IL) effects of omission cues (S-) are unique to an odor stimulus. In short, based on our current and previous observations, stimuli of different modalities can serve as effective omission cues.

We have added sentences discussing this issue in the text, which we also describe below:

While we have not tested omission cues of different modalities to suppress operant drug response, in our previous study (Suto et al., 2016, *eLife*, pii: e21920), we have successfully used auditory cues (white noise or beeping sound) as the omission cue to suppress operant lever-pressing for non-drug rewards (saccharine and glucose). Hence, the behavioral effect (response suppression) of omission cues does not appear to be unique to an odor stimulus. However, the anti-relapse action of non-olfactory omission cues against the major modes of relapse-promotion (drug availability cues, stress and drug priming) still needs to be determined in future studies. This would include examination of the relative efficacy of stimulus information presented via the primary sensory modality (olfaction in rats, vision in man) vs. secondary sensory modalities.

Regarding the neuronal effect of olfactory cues, we included the “odor control” (see Fig 3) in order to demonstrate that the neural activation in IL is due to the omission (learning) rather than olfactory (sensory) property of the S- odor (orange scent). The results (Fig 3G) indicate that the sensory property of orange scent is not sufficient to induce significant neural activation in IL (as indicated by an increase in Fos+ nuclei). Thus, the neuronal effect (activation in IL) is likely due to the learning rather than the sensory property of the S- odor. Consistent with this view, in our

previous study (Suto et al., 2016, eLife, pii: e21920), an auditory cue conditioned as the omission cue (S-) also significantly increased Fos+ nuclei in IL. Taken together, the neuronal effect of omission cues does not appear to be unique to an odor stimulus and is (at least) generalizable to an auditory cue.

3) To study the ensemble in the IL, rats are sacrificed 90 min after exposure to the orange extract sent. As pointed out by the authors, this time allows optimal induction of c-fos mRNA. However, previous literature shows that rapid (within 15min) transient changes occur in the nucleus accumbens during reinstatement and extinction. Did the authors ever look at an earlier time-point?

While this is an interesting question, we did not examine an earlier time-point. The current RNAscope study focuses on the neuronal representation of the retrieval of a pre-acquired/learned S- odor cue (i.e., memory engram) – rather than on the acquisition of learning (extinction) and the execution of behavioral response (reinstatement). Nevertheless, we found that the anti-relapse action of the S- odor cue was evident through Discrimination tests, which lasted 6.5 and 3.5 hours (from the S- onset) for cocaine and alcohol. We therefore hypothesize that the neuronal representation of the S- odor, as an anti-relapse cue, is likely similar at earlier (e.g., 15 min) and later (90 min; but perhaps even up to 6.5 hours) time points. Future work using *in vivo* calcium imaging (fiber photometry) to track IL neural activity through Discrimination training would refine the time-line and the dynamics but falls outside the scope of the current study.

4) The inactive lever presses are only reported during the first 10-14 SA sessions (Figures 1B and 2B). It would be interesting to disclose (as supplementary information if preferred) the inactive lever presses during the different post-SA stages, notably discrimination and the three types of reinstatement.

We have added three figures (Supplemental Figs 1, 2 and 4) depicting the inactive lever-presses (average \pm SEM) during the discrimination training and tests. We have also described these results in the text. In summary, the responding on inactive lever remained minimal throughout the post-SA stages.

5) In the first part of the study, rats undergo prolonged access to cocaine (2-3 weeks, 4h sessions) in order to increase compulsive intake despite adverse consequences. Did the authors ever conduct the OCIS procedure in standard short access cocaine SA? High levels of cue-, prime- and stress-induced reinstatement are observed after short access, would S- also prevent relapse then?

While this is an interesting question, we have not conducted the OCIS procedure in animals trained with standard short access (ShA) cocaine self-administration (1-2 hours daily for 2-3 weeks). We believe that the most convincing demonstration of the omission cue's efficacy is in the subjects exhibiting the most robust forms of drug seeking. As a result, we devoted our time and resource to test the anti-relapse action of cocaine omission cues in rats with extensive cocaine intake histories known to result in more robust operant drug response. Based on the fact that the anti-relapse action of cocaine S- was observed for cue-, drug priming- and stress-induced

reinstatement in rats trained with more extensive access, cocaine S- will also likely prevent drug reinstatement in ShA animals across all three major modes of relapse-promotion.

6) The cocaine-paired cue lasts for 20s, in contrast of the alcohol-paired one that lasts 5s. Is there any reason behind this large difference?

The lengths of cue-light illumination corresponded to the lengths of “time-out” periods for cocaine (20s) and alcohol (5s), during which active lever-pressing led to no scheduled consequence. The purpose of these time-outs was to prevent overdose (cocaine) and spillage/wasting (alcohol) and was adapted from our earlier studies. Based on the fact that both 20s and 5s illumination lengths reliably promoted operant lever-pressing during the No S- test (Figs 1E and 2E), the time difference between the cocaine-paired and alcohol-paired cues most likely minimally affected the behavioral outcome.

7) Daun02 disruption induced a 60% decrease in Fos+ cells (Figure 4E). Did the authors perform an RNAscope experiment to determine the neural phenotype of the 40% remaining Fos+ cells? OCIS is prevented despite the survival of this 40% of cells, do the authors hypothesize the remaining cells display the same phenotype (and this cell type can be discarded for OCIS) or do these cells form an heterogeneous pool of neurons randomly activated during the exposure to the orange extract sent, ie how do the authors explain that “surplus” of Fos+ cells?

For technical reasons, we did not perform RNAscope on the tissues after the Daun02 disruption. Daun02 disruption requires surgically placing the guide cannula and microinjection needle into IL, rendering this brain tissue fragile and unwieldy. Such tissue poses great technical challenges for RNAscope, which requires sectioning of unfixed (flush frozen) tissue into thin slices (10 μm). We thus relied on more conventional Fos immunohistochemistry (IHC), using PFA-fixed, hardened tissues sectioned into 40 μm , to validate the Daun02 disruption. In addition, due to the chemistry involved, IHC tends to be insufficiently specific for 4 or more multiplexing. Thus, it will be challenging to determine “the neural phenotype of the 40% remaining Fos+ cells” across three major cortical phenotypes (Glu/GABA/ACh), as we determined in Fig 3J, using RNAscope.

However, based on the fact that the ‘unconditioned’ orange extract scent did not significantly increase Fos+ nuclei beyond the “no odor cue” control (Fig 3G), we believe that the “surplus” of Fos+ cells, which survived the Daun02 disruption (Fig 4E), are not likely due to “the exposure to the orange extract scent” per se. Instead, such “surplus” is likely due to the basal (spontaneous) cortical activity and likely represents a different set of spontaneously active neurons than would have been spontaneously active during the Daun02 disruption.

Testing this assumption requires *in vivo* tracking of spontaneously active cells before and after the Daun02 disruption (perhaps using calcium imaging), a technically challenging undertaking. However, we would like to note that the number of the remaining Fos+ cells – after Daun02 disruption of S- reactive IL neurons – (Fig 4E: “S- & Daun02”: 25.5 ± 3.7 nuclei per mm^2) is nearly identical to the corresponding number in the “no odor cue” control (23.2 ± 5.1 nuclei per mm^2 ; see Fig 3G). Taken together, the “surplus” of Fos+ cells likely represents “a heterogeneous pool of neurons randomly activated” in wake animals.

We have discussed these considerations in the text.

Minor concerns/questions/suggestions:

1) Mention the species studied in the title.

We have added the species (rat) in the title. The revised title is “Relapse-suppression by drug omission cues: anti-relapse neurons in the infralimbic cortex of rats”.

2) In line with NIH guidelines to consider sex as a biological variable, mention earlier in the manuscript (in the introduction?) that the study was performed exclusively in males. Consider performing the key experiments (response to cocaine or alcohol availability cues for No S- and S- groups, cue-reinstatement after Daun02 disruption) in a cohort of female rats and compare to males’ response.

We agree that this is an important comment regarding sex as a biological variable. In the Introduction, we clarified that only males were tested in the current study. We received the funding for this study before the NIH guidelines to consider sex as a biological variable were implemented. Consequently, we have only received the funding to test male rats. However, we have since proposed similar projects in female rats/mice to test this important question, and we plan to compare both anti-relapse action and neuronal representation of S- across sexes in future studies.

3) Consider using the terms “cues-/drug/stress-induced reinstatement” throughout the text and figures. It is more widely found in the literature and less confusing than terms like “relapse-promotion”, “discrimination tests” or “within-session extinction”.

We agree that the suggested terms, “cues-/drug/stress-induced reinstatement”, are more commonly used in literature. However, we believe that these terms do not necessarily describe the current experimental procedures accurately.

For example, under a typical drug reinstatement model, operant drug response is “extinguished” prior to the “reinstatement” test. In contrast, under the OCIS procedure, operant drug response was not extinguished – in the absence of the S- odor – during the No S- training (Discrimination training: Figs 1C and 2C) prior to the No S- test (Discrimination tests: Figs 1E-H and 2E-G). Thus, the first time block (2 hr [cocaine] or 1 hr [alcohol]) of the No S- test was not designed to determine the “reinstatement” of “extinguished” operant response. Instead, during this time block, operant drug response – initiated and maintained by drug availability cues (active lever and light-cue) – was extinguished (“within-session” extinction) for the first time in the absence of S- (i.e., No S-). While saline or water priming, given at the start of the second time block of the No S- test (2 hrs or 1 hr into each test), failed to “reinstatement” the then “extinguished” response, cocaine or alcohol priming, given at the start of the third time block (4 hrs or 2 hours into each test) successfully reinstated drug response (“within-session” reinstatement).

Considering the above, we decided to use the more ‘descriptive’ terms, such as “relapse-promotion” and “relapse-suppression”, to articulate the opposing actions of drug availability and omission cues (and stress and drug priming). In addition, for some phrases (e.g., Discrimination tests), we prefer to keep the same terms used in our previous publications (e.g., Suto et al., 2013, *J Neurosci*, 33: 9050-5; Suto et al., 2016, *eLife*, pii: e21920) for the sake of continuity.

Nevertheless, we tried to clarify the use of these terms in this revision.

4) In the figures, consider removing “Saline SA” when presenting the discrimination tests/reinstatements. It might be understood as the data presented represents the behaviour of a control group. Stating that animals receive saline instead of cocaine infusions during reinstatements in the methods is clear enough.

We agree and have removed “Saline SA” from the figures. However, we left “cocaine” vs “saline” (Fig 1) and “alcohol” vs “water” (Fig 2) to contrast different training/test conditions.

5) It is possible that rats experience distress in presence of the orange extract scent since it is linked to withdrawal. Did the authors perform any stress/anxiety test on trained rats in presence of the scent?

We did not perform any stress/anxiety tests on trained rats in the presence of the scent. However, the reviewer raises an interesting point, and this possibility needs to be tested in future studies—especially for alcohol, which can lead to clear withdrawal symptoms. Nevertheless, we have shown that cues linked to alcohol withdrawal indeed promote – rather than suppress – operant drug response (Kufahl et al., 2011, *Neuropsychopharmacology*, 36: 2762-73). Thus, the anti-relapse action of S- to suppress operant responding (Figs 1/2/4) is likely independent of the property of the orange extract scent as a “withdrawal cue”.

6) In the alcohol part of the study, active lever pressing induced alcohol/water delivery in a well. Did the authors control for actual intake of the liquid?

We did not control for actual intake of the liquid. However, since the rats maintained higher responding for alcohol (No S- training) than water (S- training), it is reasonable to assert that the rats actually consumed alcohol throughout Discrimination training.

Reviewer #3 (Remarks to the Author):

In this manuscript Laque et al. report the results of a series of studies on the ability of cues signaling drug unavailability to suppress “relapse” to drug and alcohol seeking, and the involvement of infralimbic neurons activated by these cues in suppressing drug seeking behaviors. Overall the studies are labor intensive and represent a valuable contribution to the field. The statistical analysis is robust and appropriate, and the studies appear well-conducted and well controlled, for the most part. I do have some concerns and questions about specific aspects of the study design and control groups, that slightly attenuate my enthusiasm, described below:

Major Points:

1. The evidence presented for the importance of this specific S- activated IL ensemble in suppressing cocaine seeking is somewhat weak. Given previous work from some of the authors (Pfarr et al 2015) showing that ablation of IL neurons activated by alcohol cues results in elevated alcohol seeking behavior, how specific is this IL neural ensemble for suppression of drug and alcohol seeking? Here, the authors only look at the effects of ablated neurons activated during S- or no S-. What if the authors had ablated IL neurons activated by the cue light predicting drug delivery – might those neurons also be important for suppression of drug seeking? This would be a stronger control.

On the surface, the current study and Pfarr et al. (2015, *J Neurosci*, 35: 10750-61) provide seemingly contradictory results and interpretations regarding the behavioral function of IL neurons in the environmental control of drug relapse. However, the numerous differences in experimental design (OCIS vs. reinstatement) make straightforward comparisons of these results difficult. Nevertheless, in conjunction with our previous observations (Bossert et al., 2011, *Nat Neurosci*, 14:420-2; Warren et al., *J Neurosci*, 2016, 36: 6691-703; Suto et al., 2016, *eLife*, pii: e21920), we believe that these seemingly contradictory results can be explained by the differences in environmental contexts in which Daun02 disruption was achieved.

In the current study, Daun02 disruption of IL neurons reactive to cocaine omission cues (which inhibit behavior) was achieved in a well-habituated, behaviorally “neutral” context. In Pfarr et al. (2015), Daun02 disruption of IL neurons reactive to alcohol availability cues (which excite behavior) was achieved in an “extinguished” alcohol-predictive context (which inhibits behavior by recruiting IL neurons). These arrangements may have simultaneously disrupted two distinct units of IL neurons, each exerting opposing behavioral actions. Consistent with this presumption, Daun02 disruption of IL neurons reactive to an “extinguished” food-predictive context (which inhibits behavior) increased – rather than decreased – operant response, while Daun02 disruption of IL neurons reactive to a “non-extinguished” food-predictive context (which excites behavior) decreased operant response (Warren et al., 2016). Similarly, Daun02 disruption of IL neurons reactive to a “non-extinguished” heroin-predictive context (which excites behavior) also decreased operant response (Bossert et al., 2011). Finally, using a behavioral procedure similar to the OCIS procedure for non-drug (sugar) rewards along with the activity-based neural ablation by Daun02 in a behaviorally “neutral” context, we recently demonstrated that distinct functional units of neurons – each selectively responsible for the opposing actions of availability and omission cues on operant response – co-exist within IL (Suto et al., 2016). Taken together, it appears critical to control all environmental stimuli, including the background context for activity-based brain cell manipulations, to unambiguously establish the “cue-specificity” of brain behavioral function.

In addition, the suggested control study, which would utilize extensive behavioral procedures similar to OCIS (~5 months) and require a large number of animals but would focus on relapse-promotion, would be beyond the scope of the current study on relapse-suppression. Moreover, given the large number of differences between each study, such a control study may still not directly address the given concerns in a straightforward manner.

That said, the reviewer notes an observation that is clearly worth addressing in the manuscript. We have thus added two paragraphs addressing these issues in the Discussion section. That said, the reviewer notes an observation that is clearly worth addressing in the manuscript. We have thus added two paragraphs addressing these issues in the Discussion section.

2. In the alcohol experiments all rats appear to receive liquid alcohol diet at the same time; changes in self-administration and relapse during the later tests are then attributed to alcohol dependence and withdrawal. But the effects of additional self-administration training with alcohol cannot be ruled out as there does not appear to be a parallel control group that receives additional training without the intervening alcohol diet. Additionally, the authors don't demonstrate any dependence or withdrawal specific behavioral effects that can be linked to alcohol seeking, other than a modest increase in alcohol self-administration after the liquid diet exposure period. Given that S- cues have the same effect before and after the alcohol dependence induction, this does not seem to be a critical aspect of the study. But the authors should either modify their text to include this caveat, or include other further clarification.

While we used a regimen of liquid alcohol diet linked to alcohol dependence (Macey et al. 1996, *Alcohol*, 13: 163-170), we agree that this is an important issue that needs to be further clarified. We have added sentences pertaining to this caveat.

3. Please clarify whether the S- and no S- discrimination tests were counterbalanced for order.

The S- and No S- tests were counterbalanced between subjects. For this, rats were randomly assigned into two groups: some rats were first tested for S-, while others were first tested for No S-. We have specified this arrangement in the text.

4. Describing the initial cue availability tests as “relapse tests” seems inaccurate, as rats are not increasing responding from a pre-test baseline (lowered based on extinction or punishment or some other manipulation), but are maintaining responding from the previous discrimination based on learning. For the no S- sessions these appear to be identical to the end of discrimination training.

This comment is relevant to one of the “minor” comments (#3) from Reviewer 2 and to our corresponding response. We agree that “describing the initial cue availability test” as a “relapse tests” is not necessarily correct, and we have thus used a more descriptive term, “Discrimination tests”.

As noted, a reinstatement test requires the target operant responding to be first extinguished prior to the test; reinstatement is determined by assessing an increase in “extinguished” responding from a pre-test baseline (e.g., responding during the last extinction training). In contrast, the operant drug seeking (active lever-pressing) during the first 2 hr (cocaine) or 1 hr (alcohol) blocks of the “No S-” test (Figs 1E-H and 2E-G) – which was sufficiently initiated and maintained (promoted) by the “drug availability cues” (active lever and light-cue) – was not extinguished during the “No S-” training (Figs 1C and 2C).

However, the rats received saline/water/no-stress priming before receiving cocaine/alcohol/stress priming to establish such pre-test baselines (within-session extinction and reinstatement). The reinstating or “relapse-promoting” action of cocaine/alcohol/stress priming was suppressed by the S- cue. We thus used the perhaps more descriptive terms “anti-relapse” or “relapse-suppression” as the phrases of choice to describe the inhibitory action of S- against cocaine/alcohol/stress priming. For the sake of simplicity, we also decided to use these same terms to describe the similar inhibitory action of S- against cocaine/alcohol “availability cues”.

Nevertheless, we tried to clarify the use of these terms in this revision.

Minor Points

1. In the introduction the authors make a point of saying that their conditions are more “translationally-relevant” – could they be more explicit about the ways in which these studies are more translationally relevant than previous work?

We have clarified this section. In short, we believe that the current study is more translationally-relevant because the anti-relapse action of drug omission cues was tested against all three major modes of relapse-promotion (drug-predictive cues, stress, and drug exposure) across two major classes of abused drugs (cocaine and alcohol) under laboratory conditions linked to compulsive drug use and heightened relapse risk. Thus, the current behavioral (OCIS) procedure can be utilized to study the brain mechanisms that suppress drug relapse in drug addiction in general, thereby providing “druggable” targets for developing a new line of anti-relapse medications.

2. The description of the relapse tests in the Results section is somewhat confusing. The Methods section clarifies that each test consisted of a cue availability period, followed by saline or no shock, followed by cocaine or alcohol or shock, but this basic structure should be more clear earlier in the manuscript.

We have clarified this section.

3. How frequent were the ethanol training sessions? The methods section states that the first week of training (“5 sessions”) consisted of 14 hr training sessions, but Figure 2B only has 3 sessions at 14 hours. Were the rats trained with alcohol every other day, or were only 3/5 sessions that week 14 hrs long?

We apologize for this mistake. During the first week, the rats underwent 3 sessions (not 5 sessions) of 14-hr training over 5 days (Mon-Fri). Each 14-hr training session was conducted every other day (Mon, Wed and Fri). In contrast, the 1-hr training sessions were conducted daily.

We have corrected this mistake in this revision.

4. There is an error in the Figure 2F/G legend: only 2 conditions are listed, each twice

We apologize for this error. We have corrected it in this revision.

5. What is the purpose of the post-training habituation phase (1 wk) in Figure 3; I could not find any mention of this in the text

We have added sentences describing the purpose of this habituation phase. In this study (Fig 3), we aimed to isolate neuronal activity due specifically to S- (or habituated orange scent). During Discrimination training, each rat was subjected to alternating once daily sessions to pair “No S- and alcohol” and “S- and water (no alcohol)”. We were concerned that this routine cycle (the “No S-” training followed by the “S-” training, or vice versa) would result in neural activity due to the “expectancy” or higher basal (“background”) Fos expression independent of the S- presentation during the Discrimination test. We thus implemented the habituation phase to break the cycle. However, we do not have the data to systematically assess whether this procedure has indeed minimized background Fos expression during Daun02 disruption.

REVIEWERS' COMMENTS:

Reviewer #1 (Remarks to the Author):

I am satisfied with author's revision.

Reviewer #2 (Remarks to the Author):

The authors have positively considered all of my concerns in the previous review.

Reviewer #3 (Remarks to the Author):

The authors have satisfactorily addressed my concerns.